# Learning Rate Free Sampling in Constrained Domains

**Louis Sharrock**
Department of Mathematics and Statistics
Lancaster University, UK
l.sharrock@lancaster.ac.uk

**Lester Mackey**
Microsoft Research New England
Cambridge, MA
lmackey@microsoft.com

**Christopher Nemeth**
Department of Mathematics and Statistics
Lancaster University, UK
c.nemeth@lancaster.ac.uk

## Abstract

We introduce a suite of new particle-based algorithms for sampling in constrained domains which are entirely learning rate free. Our approach leverages coin betting ideas from convex optimisation, and the viewpoint of constrained sampling as a mirrored optimisation problem on the space of probability measures. Based on this viewpoint, we also introduce a unifying framework for several existing constrained sampling algorithms, including mirrored Langevin dynamics and mirrored Stein variational gradient descent. We demonstrate the performance of our algorithms on a range of numerical examples, including sampling from targets on the simplex, sampling with fairness constraints, and constrained sampling problems in post-selection inference. Our results indicate that our algorithms achieve competitive performance with existing constrained sampling methods, without the need to tune any hyperparameters.

## 1 Introduction

The problem of sampling from unnormalised probability distributions is of central importance to computational statistics and machine learning. Standard approaches to this problem include Markov chain Monte Carlo (MCMC) [11, 76] and variational inference (VI) [41, 96]. More recently, there has been growing interest in particle-based variational inference (ParVI) methods, which deterministically evolve a collection of particles towards the target distribution and combine favourable aspects of both MCMC and VI. Perhaps the most well-known of these methods is Stein variational gradient descent (SVGD) [63], which iteratively updates particles according to a form of gradient descent on the Kullback-Leibler (KL) divergence. This approach has since given rise to several extensions [16, 27, 97, 105] and found success in a wide range of applications [27, 66, 74].

While such methods have enjoyed great success in sampling from unconstrained distributions, they typically break down when applied to constrained targets [see, e.g., 60, 82]. This is required for targets that are not integrable over the entire Euclidean space (e.g., the uniform distribution), when the target density is undefined outside a particular domain (e.g., the Dirichlet distribution), or when the target must satisfy certain additional constraints (e.g., fairness constraints in Bayesian inference [65]). Notable examples include latent Dirichlet allocation [9], ordinal data models [45], regularised regression [15], survival analysis [49], and post-selection inference [54, 88]. In recent years, there has been increased interest in this topic, with several new methodological and theoretical developments [1, 19, 43, 44, 60, 61, 82, 86, 103, 104].

37th Conference on Neural Information Processing Systems (NeurIPS 2023).

One limitation shared by all of these methods is that their performance depends, often significantly, on a suitable choice of learning rate. In principle, convergence rates for existing constrained sampling schemes (e.g., [1, Theorem 1] or [86, Corollary 1]) allow one to compute the optimal learning rate for a given problem. In practice, however, this optimal learning rate is a function of the unknown target and thus cannot be computed. Motivated by this problem, in this paper we introduce a suite of new sampling algorithms for constrained domains which are entirely learning rate free.

**Our contributions** We first propose a unifying perspective of several existing constrained sampling algorithms, based on the viewpoint of sampling in constrained domains as a 'mirrored' optimisation problem on the space of probability measures. Based on this perspective, we derive a general class of solutions to the constrained sampling problem via the mirrored Wasserstein gradient flow (MWGF). The MWGF includes, as special cases, several existing constrained sampling methods, e.g., mirrored Langevin dynamics [43] and mirrored SVGD (MSVGD) [82]. Using this formulation, we also introduce mirrored versions of several other particle-based sampling algorithms which are suitable for sampling in constrained domains, and study their convergence properties in continuous time.

By leveraging this perspective, and extending the coin sampling methodology recently introduced in [81], we then obtain a suite of learning rate free algorithms suitable for constrained domains. In particular, we introduce mirrored coin sampling, which includes as a special case Coin MSVGD, as well as several other tuning-free constrained sampling algorithms. We also outline an alternative approach which does not rely on a mirror mapping, namely, coin mollified interaction energy descent (Coin MIED). This algorithm can be viewed as the natural coin sampling analogue of the method recently introduced in [60]. Finally, we demonstrate the performance of our methods on a wide range of examples, including sampling on the simplex, sampling with fairness constraints, and constrained sampling problems in post-selection inference. Our empirical results indicate that our methods achieve competitive performance with the optimal performance of existing approaches, with no need to tune a learning rate.

## 2 Preliminaries

**The Wasserstein Space**. We will make use of the following notation. For any $\mathcal{X} \subset \mathbb{R}^d$, we write $\mathcal{P}_2(\mathcal{X})$ for the set of probability measures $\mu$ on $\mathcal{X}$ with finite $2^{\text{nd}}$ moment: $\int_{\mathcal{X}} ||x||^2 \mu(\mathrm{d}x) < \infty$, where $||\cdot||$ denotes the Euclidean norm. For $\mu \in \mathcal{P}_2(\mathcal{X})$, we write $L^2(\mu)$ for the set of measurable $f : \mathcal{X} \to \mathcal{X}$ with finite $||f||^2_{L^2(\mu)} = \int_{\mathcal{X}} ||f(x)||^2 \mu(\mathrm{d}x)$. For $f, g \in L^2(\mu)$, we also define $\langle f, g \rangle_{L^2(\mu)} = \int_{\mathcal{X}} f(x)^\top g(x) \mu(\mathrm{d}x)$. Given $\mu \in \mathcal{P}_2(\mathcal{X})$ and $T \in L^2(\mu)$, we write $T_\# \mu$ for the pushforward of $\mu$ under $T$, that is, the distribution of $T(X)$ when $X \sim \mu$.

Given $\mu, \nu \in \mathcal{P}_2(\mathcal{X})$, the Wasserstein 2-distance between $\mu$ and $\nu$ is defined according to $W_2^2(\mu, \nu) = \inf_{\gamma \in \Gamma(\mu, \nu)} \int_{\mathcal{X} \times \mathcal{X}} ||x - y||^2 \gamma(\mathrm{d}x, \mathrm{d}y)$, where $\Gamma(\mu, \nu)$ denotes the set of couplings between $\mu$ and $\nu$. The Wasserstein distance $W_2$ is indeed a distance over $\mathcal{P}_2(\mathcal{X})$ [2, Chapter 7.1], with the metric space $(\mathcal{P}_2(\mathcal{X}), W_2)$ known as the Wasserstein space. Given a functional $\mathcal{F} : \mathcal{P}(\mathcal{X}) \to (-\infty, \infty]$, we will write $\nabla_{W_2} \mathcal{F}(\mu)$ for the Wasserstein gradient of $\mathcal{F}$ at $\mu$, which exists under appropriate regularity conditions [2, Lemma 10.4.1]. We refer to App. A for further details on calculus in the Wasserstein space.

**Wasserstein Gradient Flows**. Suppose that $\pi \in \mathcal{P}_2(\mathcal{X})$ and that $\pi$ admits a density $\pi \propto e^{-V}$ with respect to Lebesgue measure,[1] where $V : \mathcal{X} \to \mathbb{R}$ is a smooth potential function. The problem of sampling from $\pi$ can be recast as an optimisation problem over $\mathcal{P}_2(\mathcal{X})$ [e.g., 99]. In particular, one can view the target $\pi$ as the solution of

$$\pi = \underset{\mu \in \mathcal{P}_2(\mathcal{X})}{\arg\min} \mathcal{F}(\mu) \tag{1}$$

where $\mathcal{F} : \mathcal{P}(\mathcal{X}) \to (-\infty, \infty]$ is a functional uniquely minimised at $\pi$. In the unconstrained case, $\mathcal{X} = \mathbb{R}^d$, a now rather classical method for solving this problem is to identify a continuous process which pushes samples from an initial distribution $\mu_0$ to the target $\pi$. This corresponds to finding a family of vector fields $(v_t)_{t \geq 0}$, with $v_t \in L^2(\mu_t)$, which transports $\mu_t$ to $\pi$ via the continuity equation

$$\frac{\partial \mu_t}{\partial t} + \nabla \cdot (v_t \mu_t) = 0. \tag{2}$$

---

[1]In a slight abuse of notation, throughout this paper we will use the same letter to refer to a distribution and its density with respect to Lebesgue measure.

A standard choice is $v_t = -\nabla_{W_2}\mathcal{F}(\mu_t)$, in which case the solution $(\mu_t)_{t\geq 0}$ of (2) is referred to as the Wasserstein gradient flow (WGF) of $\mathcal{F}$. In fact, for different choices of $(v_t)_{t\geq 0}$, one can obtain the continuous-time limit of several popular sampling algorithms [3, 20, 40, 46, 51, 63, 99]. Unfortunately, it is not straightforward to extend these approaches to the case in which $\mathcal{X} \subset \mathbb{R}^d$ is a constrained domain. For example, the vector fields $v_t$ may push the random variable $x_t$ outside of its support $\mathcal{X}$, thus rendering all future updates undefined.

## 3 Mirrored Wasserstein Gradient Flows

We now outline one way to extend the WGF approach to the constrained case, based on the use of a mirror map. We will first require some basic definitions from convex optimisation [e.g., 77]. Let $\mathcal{X}$ be a closed, convex domain in $\mathbb{R}^d$. Let $\phi : \mathcal{X} \to \mathbb{R} \cup \{\infty\}$ be a proper, lower semicontinuous, strongly convex function of Legendre type [77]. This implies, in particular, that $\nabla\phi(\mathcal{X}) = \mathbb{R}^d$ and $\nabla\phi : \mathcal{X} \to \mathbb{R}^d$ is bijective [77, Theorem 26.5]. Moreover, its inverse $(\nabla\phi)^{-1} : \mathbb{R}^d \to \mathcal{X}$ satisfies $(\nabla\phi)^{-1} = \nabla\phi^*$, where $\phi^* : \mathbb{R}^d \to \mathbb{R}$ denotes the Fenchel conjugate of $\phi$, defined as $\phi^*(y) = \sup_{x \in \mathcal{X}} \langle x, y \rangle - \phi(x)$. We will refer to $\nabla\phi : \mathcal{X} \to \mathbb{R}^d$ as the mirror map and $\nabla\phi(\mathcal{X}) = \mathbb{R}^d$ as the dual space.

### 3.1 Constrained Sampling as Optimisation

Using the mirror map $\nabla\phi : \mathcal{X} \to \mathbb{R}^d$, we can now reformulate the constrained sampling problem as the solution of a 'mirrored' version of the optimisation problem in (1). Let us define $\nu = (\nabla\phi)_{\#}\pi$, with $\pi = (\nabla\phi^*)_{\#}\nu$. We can then view the target $\pi$ as the solution of

$$\pi = (\nabla\phi^*)_{\#}\nu, \quad \nu = \underset{\eta \in \mathcal{P}_2(\mathbb{R}^d)}{\arg\min} \mathcal{F}(\eta), \tag{3}$$

where now $\mathcal{F} : \mathcal{P}_2(\mathbb{R}^d) \to (-\infty, \infty]$ is a functional uniquely minimised by $\nu = (\nabla\phi)_{\#}\pi$. The motivation for this formulation is clear. Rather than directly solving a constrained optimisation problem for the target $\pi \in \mathcal{P}_2(\mathcal{X})$ as in (1), we could instead solve an unconstrained optimisation problem for the mirrored target $\nu \in \mathcal{P}_2(\mathbb{R}^d)$, and then recover the target via $\pi = (\nabla\phi^*)_{\#}\nu$. This is a rather natural extension of the formulation of sampling as optimisation to the constrained setting.

### 3.2 Mirrored Wasserstein Gradient Flows

One way to solve the mirrored optimisation problem in (3) is to find a continuous process which transports samples from $\eta_0 := (\nabla\phi)_{\#}\mu_0 \in \mathcal{P}_2(\mathbb{R}^d)$ to the mirrored target $\nu$. According to the mirror map, this process will implicitly also push samples from $\mu_0$ to the target $\pi$. More precisely, we would like to find $(w_t)_{t\geq 0}$, where $w_t : \mathbb{R}^d \to \mathbb{R}^d$, which evolves $\eta_t$ on $\mathbb{R}^d$ and thus $\mu_t$ on $\mathcal{X}$, through

$$\frac{\partial \eta_t}{\partial t} + \nabla \cdot (w_t \eta_t) = 0 \quad, \quad \mu_t = (\nabla\phi^*)_{\#}\eta_t \tag{4}$$

where $(w_t)_{t\geq 0}$ should ensure convergence of $\eta_t$ to $\nu$. In the case that $w_t = -\nabla_{W_2}\mathcal{F}(\eta_t)$, we will refer to (4) as the mirrored Wasserstein gradient flow (MWGF) of $\mathcal{F}$. The idea of (4) is to simulate the WGF in the (unconstrained) dual space and then map back to the primal (constrained) space using a carefully chosen mirror map. This approach bypasses the inherent difficulties associated with simulating a gradient flow on the constrained space itself.

While a similar approach has previously been used to derive certain mirrored sampling schemes [43, 82, 86], these works focus on specific choices of $\mathcal{F}$ and $(w_t)_{t\geq 0}$, while we consider the general case. Our general perspective provides a unifying framework for these approaches, whilst also allowing us to obtain new constrained sampling algorithms and to study their convergence properties (see App. B). We now consider one such approach in detail.

### 3.2.1 Mirrored Stein Variational Gradient Descent

MSVGD [82] is a mirrored version of the popular SVGD algorithm. In [82], MSVGD is derived using a mirrored Stein operator in a manner analogous to the original SVGD derivation [63]. Here, we take an alternative perspective, which originates in the MWGF. This formulation is more closely aligned with other more recent papers analysing the non-asymptotic convergence of SVGD [51, 78, 85, 86].

---

**Algorithm 1** MSVGD

---

**input:** target density $\pi$, kernel $k$, mirror function $\phi$, particles $(x_0^i)_{i=1}^N \sim \mu_0$, step size $\gamma$.
**initialise:** $y_0^i = \nabla\phi(x_0^i)$ for $i \in [N]$
**for** $t = 0, \dots, T-1$ **do**
    For $i \in [N], y_{t+1}^i = y_t^i - \gamma P_{\hat{\eta}_t, k_\phi} \nabla \log\left(\frac{\hat{\eta}_t}{\nu}\right)(y_t^i)$, where $\hat{\eta}_t = \frac{1}{N}\sum_{j=1}^N \delta_{y_t^j}$.
    For $i \in [N], x_{t+1}^i = \nabla\phi^*(y_{t+1}^i)$.
**return** $(x_T^i)_{i=1}^N$.

---

We will require the following additional notation. Let $k : \mathcal{X} \times \mathcal{X} \to \mathbb{R}$ denote a positive semi-definite kernel and $\mathcal{H}_k$ the corresponding reproducing kernel Hilbert space (RKHS) of real-valued functions $f : \mathcal{X} \to \mathbb{R}$, with inner product $\langle \cdot, \cdot \rangle_{\mathcal{H}_k}$ and norm $|| \cdot ||_{\mathcal{H}_k}$. Let $\mathcal{H} := \mathcal{H}_k^d$ denote the Cartesian product RKHS, consisting of elements $f = (f_1, \dots, f_d)$ for $f_i \in \mathcal{H}_k$. We write $S_{\mu,k} : L^2(\mu) \to \mathcal{H}$ for the operator $S_{\mu,k}f = \int k(x, \cdot)f(x)\mathrm{d}\mu(x)$. We assume $\int k(x,x)\mathrm{d}\mu(x) < \infty$ for $\mu \in \mathcal{P}_2(\mathcal{X})$, in which case $\mathcal{H} \subset L^2(\mu)$ for all $\mu \in \mathcal{P}_2(\mathcal{X})$ [52, Sec. 3.1]. We also define the identity embedding $i : \mathcal{H} \to L^2(\mu)$ with adjoint $i^* = S_{\mu,k}$. Finally, we define $P_{\mu,k} : L^2(\mu) \to L^2(\mu)$ as $P_{\mu,k} = iS_{\mu,k}$.

We are now ready to introduce our perspective on MSVGD. We begin by introducing the continuous-time MSVGD dynamics, which are defined by the continuity equation

$$\frac{\partial\eta_t}{\partial t} - \nabla \cdot \left(P_{\eta_t, k_\phi}\nabla\log\left(\frac{\eta_t}{\nu}\right)\eta_t\right) = 0, \quad \mu_t = (\nabla\phi^*)_\#\eta_t, \tag{5}$$

where $k_\phi(y, y') = k(\nabla\phi^*(y), \nabla\phi^*(y'))$ for some base kernel $k : \mathcal{X} \times \mathcal{X} \to \mathbb{R}$. Clearly, this is a special case of the MWGF in (4), in which $(w_t)_{t\geq 0}$ are defined according to $w_t = -P_{\eta_t, k_\phi}\nabla_{W_2}\mathcal{F}(\eta_t)$, with $\mathcal{F}(\eta_t) = \mathrm{KL}(\eta_t || \nu)$. In particular, $w_t$ can be interpreted as the negative Wasserstein gradient of $\mathrm{KL}(\eta_t || \nu)$, the KL divergence of the $\eta_t$ with respect to $\nu$, under the inner product of $\mathcal{H}_{k_\phi}$.

The key point here is that, given samples from $\eta_t$, estimating $P_{\eta_t, k_\phi}\nabla\log(\frac{\eta_t}{\nu})$ is straightforward. In particular, if $\lim_{||y'||\to\infty} k_\phi(y', \cdot)\nu(y') = 0$, then, integrating by parts,

$$P_{\eta_t, k_\phi}\nabla\log(\frac{\eta_t}{\nu})(y) = -\int_{\mathbb{R}^d}[k_\phi(y', y)\nabla_{y'}\log\nu(y') + \nabla_{y'}k_\phi(y', y)]\mathrm{d}\eta_t(y'). \tag{6}$$

To obtain an implementable algorithm, we must discretise in time. By applying an explicit Euler discretisation to (5), we arrive at

$$\eta_{t+1} = \left(\mathrm{id} - \gamma P_{\eta_t, k_\phi}\nabla\log\left(\frac{\eta_t}{\nu}\right)\right)_\#\eta_t, \quad \mu_{t+1} = (\nabla\phi^*)_\#\eta_{t+1}, \tag{7}$$

where $\gamma > 0$ is a step size or learning rate, and $\mathrm{id}$ is the identity map. This is the population limit of the MSVGD algorithm in [82]. Now, suppose $\mu_0 \in \mathcal{P}_2(\mathcal{X})$ admits a density with respect to the Lebesgue measure. In addition, suppose $x_0 \sim \mu_0$ and $y_0 = \nabla\phi(x_0)$. Then, for $t \in \mathbb{N}_0$, if we define

$$y_{t+1} = y_t - \gamma P_{\eta_t, k_\phi}\nabla\log\left(\frac{\eta_t}{\nu}\right)(y_t), \quad x_{t+1} = \nabla\phi^*(y_{t+1}), \tag{8}$$

then $(\eta_t)_{t\in\mathbb{N}_0}$ and $(\mu_t)_{t\in\mathbb{N}_0}$ in (7) are precisely the densities of $(y_t)_{t\in\mathbb{N}_0}$ and $(x_t)_{t\in\mathbb{N}_0}$ in (8). In practice, these densities are unknown and must be estimated using samples. In particular, to approximate (8), we initialise a collection of particles $x_0^i \overset{\text{i.i.d.}}{\sim} \mu_0$ for $i \in [N]$, set $y_0^i = \nabla\phi(x_0^i)$, and then update

$$y_{t+1}^i = y_t^i - \gamma P_{\hat{\eta}_t, k_\phi}\nabla\log\left(\frac{\hat{\eta}_t}{\nu}\right)(y_t^i), \quad x_{t+1}^i = \nabla\phi^*(y_{t+1}^i) \tag{9}$$

where $\hat{\eta}_t = \frac{1}{N}\sum_{j=1}^N \delta_{y_t^j}$. This is precisely the MSVGD algorithm introduced in [82] and since also studied in [86].

### 3.2.2 Other Algorithms

In App. B, we outline several algorithms which arise as special cases of the MWGF. These include existing algorithms, such as the mirrored Langevin dynamics (MLD) [43] and MSVGD [82], and two new algorithms, which correspond to mirrored versions of Laplacian adjusted Wasserstein gradient descent (LAWGD) [20] and kernel Stein discrepancy descent (KSDD) [51]. We also study the convergence properties of these algorithms in continuous time.

# 4 Mirrored Coin Sampling

By construction, any algorithm derived as a discretisation of the MWGF, including MSVGD, will depend on an appropriate choice of learning rate $\gamma$. In this section, we consider an alternative approach, leading to new algorithms which are entirely learning rate free. Our approach can be viewed as an extension of the coin sampling framework introduced in [81] to the constrained setting, based on the reformulation of the constrained sampling problem as a mirrored optimisation problem.

## 4.1 Coin Sampling

We begin by reviewing the general coin betting framework [22, 71, 72]. Consider a gambler with initial wealth $w_0 = 1$ who makes bets on a series of adversarial coin flips $c_t \in \{-1, 1\}$, where $+1$ denotes heads and $-1$ denotes tails. We encode the gambler's bet by $x_t \in \mathbb{R}$. In particular, $\text{sign}(x_t) \in \{-1, 1\}$ denotes whether the bet is on heads or tails, and $|x_t| \in \mathbb{R}$ denotes the size of the bet. Thus, in the $t^{\text{th}}$ round, the gambler wins $|x_t c_t|$ if $\text{sign}(c_t) = \text{sign}(x_t)$ and loses $|x_t c_t|$ otherwise. Finally, we write $w_t$ for the wealth of the gambler at the end of the $t^{\text{th}}$ round. The gambler's wealth thus accumulates as $w_t = 1 + \sum_{s=1}^{t} c_s x_s$. We assume that the gambler's bets satisfy $x_t = \beta_t w_{t-1}$, for a betting fraction $\beta_t \in [-1, 1]$, which means the gambler does not borrow any money.

In [71], the authors show how this approach can be used to solve convex optimisation problems on $\mathbb{R}^d$, that is, problems of the form $x^* = \arg\min_{x \in \mathbb{R}^d} f(x)$, for convex $f : \mathbb{R}^d \to \mathbb{R}$. In particular, [71] consider a betting game with 'outcomes' $c_t = -\nabla f(x_t)$, replacing scalar multiplications $c_t x_t$ by scalar products $\langle c_t, x_t \rangle$. In this case, under certain assumptions, they show that $f(\frac{1}{T} \sum_{t=1}^{T} x_t) \to f(x^*)$ at a rate determined by the betting strategy. There are many suitable choices for the betting strategy. Perhaps the simplest of these is $\beta_t = t^{-1} \sum_{s=1}^{t-1} c_s$, known as the Krichevsky-Trofimov (KT) betting strategy [53, 71], which yields the sequence of bets $x_t = \beta_t w_{t-1} = t^{-1} \sum_{s=1}^{t-1} c_s (1 + \sum_{s=1}^{t-1} \langle c_s, x_s \rangle)$. Other choices, however, are also possible [22, 70, 72].

In [81], this approach was extended to solve optimisation problems on the space of probability measures. In this case, several further modifications are necessary. First, in each round, one now bets $x_t - x_0$ rather than $x_t$, where $x_0 \sim \mu_0$ is distributed according to some $\mu_0 \in \mathcal{P}_2(\mathbb{R}^d)$. In this case, viewing $x_t : \mathbb{R}^d \to \mathbb{R}^d$ as a function that maps $x_0 \mapsto x_t(x_0)$, one can define a sequence of distributions $(\mu_t)_{t \in \mathbb{N}}$ as the push-forwards of $\mu_0$ under the functions $(x_t)_{t \in \mathbb{N}}$. In particular, this means that, if $x_0 \sim \mu_0$, then $x_t \sim \mu_t$.

By using these modifications in the original coin betting framework, and choosing $(c_t)_{t \in \mathbb{N}} := (c_{\mu_t}(x_t))_{t \in \mathbb{N}}$ appropriately, [81] show that it is now possible to solve problems of the form $\mu^* = \arg\min_{\mu \in \mathcal{P}_2(\mathcal{X})} \mathcal{F}(\mu)$. This approach is referred to as coin sampling. For example, by considering a betting game with $c_t = -\nabla_{W_2} \mathcal{F}(\mu_t)(x_t)$, [81] obtain a 'coin Wasserstein gradient descent' algorithm, which can be viewed as a learning-rate free analogue of Wasserstein gradient descent. Alternatively, setting $c_t = -P_{\mu_t, k} \nabla_{W_2} \mathcal{F}(\mu_t)(x_t)$, one can obtain a learning-rate free analogue of the population limit of SVGD, in which the updates are given by

$$x_t = x_0 - \frac{\sum_{s=1}^{t-1} \mathcal{P}_{\mu_s, k} \nabla_{W_2} \mathcal{F}(\mu_s)(x_s)}{t} \left( 1 - \sum_{s=1}^{t-1} \langle \mathcal{P}_{\mu_s, k} \nabla_{W_2} \mathcal{F}(\mu_s)(x_s), x_s - x_0 \rangle \right), \quad (10)$$

where, as described above, $\mu_t = (x_t)_{\#} \mu_0$. In practice, the densities $(\mu_t)_{t \in \mathbb{N}_0}$ and thus the outcomes $(c_t)_{t \in \mathbb{N}}$ are almost always intractable, and will be approximated using a set of interacting particles.

## 4.2 Mirrored Coin Sampling

By leveraging the formulation of constrained sampling as a 'mirrored' optimisation problem over the space of probability measures, we can extend the coin sampling approach to constrained domains. In particular, as an alternative to discretising the MWGF to solve this optimisation problem (Sec. 3.2), we propose instead to use a mirrored version of coin sampling (Sec. 4.1). The idea is to use coin sampling to solve the optimisation problem in the dual space, and then map back to the primal space using the mirror map. In particular, this suggests the following scheme. Let $x_0 \sim \mu_0 \in \mathcal{P}_2(\mathcal{X})$, and $y_0 = \nabla\phi(x_0) \in \mathbb{R}^d$. Then, for $t \in \mathbb{N}$, update

$$y_t = y_0 + \frac{\sum_{s=1}^{t-1} c_{\eta_s}(y_s)}{t} \left( 1 + \sum_{s=1}^{t-1} \langle c_{\eta_s}(y_s), y_s - y_0 \rangle \right) \quad, \quad x_t = \nabla\phi^*(x_t), \quad (11)$$

---

**Algorithm 2** Coin MSVGD

---

**input:** target density $\pi$, kernel $k$, mirror function $\phi$, particles $(x_0^i)_{i=1}^N \sim \mu_0$.
**intialise:** $y_0^i = \nabla\phi(x_0^i)$ for $i \in [N]$
**for** $t = 1, \ldots, T$ **do**
    For $i \in [N], y_t^i = y_0^i - \frac{\sum_{s=1}^{t-1} P_{\hat{\eta}_s, k_\phi} \nabla \log\left(\frac{\hat{\eta}_s}{\nu}\right)(y_s^i)}{t} \left( 1 - \sum_{s=1}^{t-1} \langle P_{\hat{\eta}_s, k_\phi} \nabla \log(\frac{\hat{\eta}_s}{\nu})(y_s^i), y_s^i - y_0^i \rangle \right)$.
    For $i \in [N], x_t^i = \nabla\phi^*(y_t^i)$.
**return** $(x_T^i)_{i=1}^N$.

---

where $\eta_t = (y_t)_\# \eta_0$ and $\mu_t = (\nabla\phi^*)_\# \eta_t$, and where $c_{\eta_s}(y_s) = -\nabla_{W_2} \mathcal{F}(\eta_s)(y_s)$, or some variant thereof. We will refer to this approach as mirrored coin sampling, or mirrored coin Wasserstein gradient descent. In order to obtain an implementable algorithm, we proceed as follows. First, decide on a suitable functional $\mathcal{F}$ and a suitable $(c_t)_{t \in \mathbb{N}} := (c_{\eta_t}(y_t))_{t \in \mathbb{N}}$. For example, $\mathcal{F}(\eta) = \mathrm{KL}(\eta|\nu)$, and $(c_t)_{t \in \mathbb{N}} = (-\mathcal{P}_{\eta_t, k_\phi} \nabla_{W_2} \mathcal{F}(\eta_t)(y_t))_{t \in \mathbb{N}}$. Then, substitute these into (11), and approximate the updates using a set of interacting particles, as in existing ParVIs algorithms.

Following these steps, we obtain a learning rate free analogue of MSVGD [82], which we will refer to as Coin MSVGD. This is summarised in Alg. 2. For different choices of $(c_{\eta_t})_{t \in \mathbb{N}}$ in (11), we also obtain learning rate free analogues of two other mirrored ParVI algorithms introduced in App. B, namely mirrored LAWGD (App. B.3), and mirrored KSDD (App. B.4). These algorithms, which we term Coin MLAWGD and Coin MKSDD, are given in Alg. 6 and Alg. 7 in App. D.

Even in the unconstrained setting, establishing the convergence of coin sampling remains an open problem. In [81], the authors provide a technical sufficient condition under which it is possible to establish convergence to the target measure in the population limit; however, it is difficult to verify this condition in general. In the interest of completeness, in App. C, we provide an analogous convergence result for mirrored coin sampling, adapted appropriately to the constrained setting.

### 4.3 Alternative Approaches

In this section, we outline an alternative approach, based on a 'coinification' of the mollified interaction energy descent (MIED) method recently introduced in [60]. MIED is based on minimising a function known as the mollified interaction energy (MIE) $\mathcal{E}_\epsilon : \mathcal{P}(\mathbb{R}^d) \to [0, \infty]$. The idea is that minimising this function balances two forces: a repulsive force which ensures the measure is well spread, and an attractive force which ensures the measure is concentrated around regions of high density. In order to obtain a practical sampling algorithm, [60] propose to minimise a discrete version of the logarithmic MIE, using, e.g., Adagrad [31] or Adam [48]. That is, minimising the function

$$\log E_\epsilon(\omega^n) := \log \frac{1}{N^2} \sum_{i=1}^n \sum_{j=1}^n \phi_\epsilon(x^i - x^j)(\pi(x^i)\pi(x^j))^{-\frac{1}{2}}, \tag{12}$$

where $\omega^n = \{x^1, \ldots, x^n\}$, and where $(\phi_\epsilon)_{\epsilon > 0}$ is a family of mollifiers [60, Definition 3.1]. This approach can be adapted to handle the constrained case, as outlined in [60]. In particular, if there exists a differentiable $f : \mathbb{R}^d \to \mathcal{X}$ such that $\mathcal{X} \setminus f(\mathbb{R}^d)$ has Lebesgue measure zero, then one can reduce the constrained problem to an unconstrained one by minimising $\log E_\epsilon(f(w^n))$, where $f(\omega^n) = \{f(x^1), \ldots, f(x^n)\}$. Meanwhile, if $\mathcal{X} = \{x \in \mathbb{R}^d : g(x) \leq 0\}$ for some differentiable $g : \mathbb{R}^d \to \mathbb{R}$, then one can use a variant of the dynamic barrier method introduced in [37].

Here, as an alternative to the optimisation methods considered in [60], we instead propose to use a learning-rate free algorithm to minimise (12), based on the coin betting framework described in 4.1. We term the resulting method, summarised in Alg. 3, Coin MIED. In comparison to our mirrored coin sampling algorithm, this approach has the advantage that it only requires a differentiable (not necessarily bijective) map.

## 5 Related Work

**Sampling as Optimisation**. The connection between sampling and optimisation has a long history, dating back to [46], which established that the evolution of the law of the Langevin diffusion

**Algorithm 3** Coin MIED

---

**Input:** target density $\pi$, mollifier $\phi_\epsilon$, particles $\omega_0^n = (x_0^i)_{i=1}^N$.
**for** $t = 1, \ldots, T$ **do**

For $i \in [N], x_t^i = x_0^i - \frac{\sum_{s=1}^{t-1} \nabla_{x_s^i} \log E_\epsilon(\omega_s^n)}{t} \left( 1 - \sum_{s=1}^{t-1} \langle \nabla_{x_s^i} \log E_\epsilon(\omega_s^n), x_s^i - x_0^i \rangle \right)$.

**return** $(x_T^i)_{i=1}^N$.

---

corresponds to the WGF of the KL divergence. In recent years, there has been renewed interest in this perspective [99]. In particular, the viewpoint of existing algorithms such as LMC [23, 24, 33, 34] and SVGD [32, 63] as discretisations of WGFs has proved fruitful in deriving sharp convergence rates [6, 34, 52, 78, 83, 85]. It has also resulted in the rapid development of new sampling algorithms, inspired by ideas from the optimisation literature such as proximal methods [100], coordinate descent [29, 30], Nesterov acceleration [18, 25, 67], Newton methods [68, 84, 98], and coin betting [80, 81].

**Sampling in Constrained Domains**. Recent interest in constrained sampling has resulted in a range of general-purpose algorithms for this task [e.g., 43, 60, 82, 103, 104]. In cases where it is possible to explicitly parameterise the target domain in a lower dimensional space, one can use variants of classical methods, including LMC [98], rejection sampling [28], Hamiltonian Monte Carlo (HMC) [13], and Riemannian manifold HMC [55, 73]. Several other approaches are based on the use of mirror map. In particular, [43] proposed mirrored Langevin dynamics and its first-order discretisation. [103], and an earlier draft of [43], introduced the closely related mirror Langevin diffusion and the mirror Langevin algorithm, which has since also been studied in [1, 19, 44, 61]. Along the same lines, [82] introduced two variants of SVGD suitable for constrained domains based on the use of a mirror map, and established convergence of these schemes to the target distribution; see also [86]. In certain cases [e.g., 56, 58] one cannot explicitly parameterise the manifold of interest. In this setting, different approaches are required. [12] introduced a constrained version of HMC for this case. Meanwhile, [102] proposed a constrained Metropolis-Hastings algorithm with a reverse projection check to ensure reversibility; this approach has since been extended in [57, 59]. Other, more recent approaches suitable for this setting include MIED [60] (see Sec. 4.3), and O-SVGD [104].

## 6  Numerical Results

In this section, we perform an empirical evaluation of Coin MSVGD (Sec. 6.1 - 6.2) and Coin MIED (Sec. 6.3). We consider several simulated and real data experiments, including sampling from targets defined on the simplex (Sec. 6.1), confidence interval construction for post-selection inference (Sec. 6.2), and inference in a fairness Bayesian neural network (Sec. 6.3). We compare our methods to their learning-rate-dependent analogues, namely, MSVGD [82] and MIED [60]. We also include a comparison with Stein Variational Mirror Descent (SVMD) [82], and projected versions of Coin SVGD [81] and SVGD [63], which include a Euclidean projection step to ensure the iterates remain in the domain. We provide additional experimental details in App. F and additional numerical results in App. G. Code to reproduce all of the numerical results can be found at https://github.com/louissharrock/constrained-coin-sampling.

### 6.1  Simplex Targets

Following [82], we first test the performance of our algorithms on two 20-dimensional targets defined on the simplex: the sparse Dirichlet posterior of [73] and the quadratic simplex target of [1]. We employ the IMQ kernel and the entropic mirror map [7]; and use $N = 50$ particles, $T = 500$ iterations. In Fig. 1 and Fig. 6 - 7 (App. F.1), we plot the energy distance [87] to a set of surrogate ground truth samples, either obtained i.i.d. (sparse Dirichlet target) or using the No-U-Turn Sampler (NUTS) [42] (quadratic target). After $T = 500$ iterations, Coin MSVGD has comparable performance to MSVGD and SVMD with optimal learning rates but significantly outperforms both algorithms for sub-optimal learning rates (Fig. 1). In both cases, the projected methods fail to converge. For the sparse Dirichlet posterior, Coin MSVGD converges more rapidly than MSVGD and SVMD, even for well-chosen values of the learning rate (Fig. 6a in App. F.1). Meanwhile, for the quadratic target, Coin MSVGD generally converges more rapidly than MSVGD but not as fast as SVMD [82], which takes advantage of the log-concavity of the target in the primal space (Fig. 6a in App. F.1).

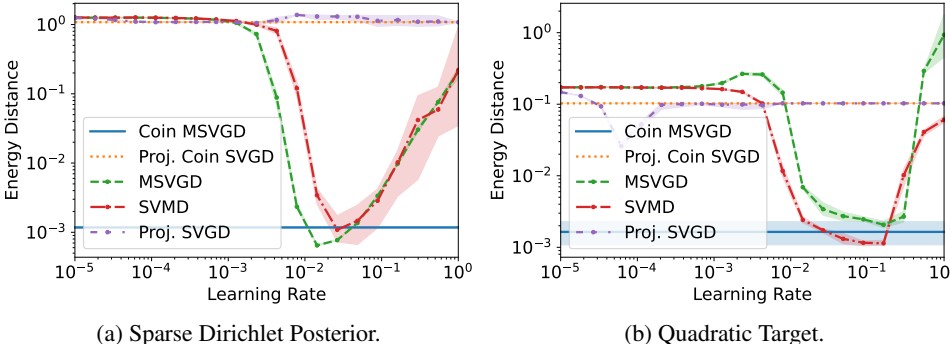

(a) Sparse Dirichlet Posterior.

(b) Quadratic Target.

Figure 1: **Results for the simplex targets in [73] and [1]**. Posterior approximation quality for Coin MSVGD, MSVGD, projected Coin SVGD, projected SVGD, and SVMD.

## 6.2 Confidence Intervals for Post Selection Inference

We next consider a constrained sampling problem arising in post-selection inference [54, 88]. Suppose we are given data $(X, y) \in \mathbb{R}^{n \times p} \times \mathbb{R}^n$. We are interested in obtaining valid confidence intervals (CIs) for regression coefficients obtained via the randomised Lasso [89], defined as the solution of

$$\hat{\beta} = \arg\min_{\beta \in \mathbb{R}^p} \left[ \frac{1}{2} ||y - X\beta||_2^2 + \lambda ||\beta||_1 - \omega^\top \beta + \frac{\varepsilon}{2} ||\beta||_2^2 \right], \tag{13}$$

where $\lambda, \varepsilon \in \mathbb{R}_+$ are penalty parameters and $\omega \in \mathbb{R}^p$ is an auxiliary random vector. Let $\hat{\beta}_E \in \mathbb{R}^q$ represent the non-zero coefficients of $\hat{\beta}$ and $\hat{z}_E = \text{sign}(\hat{\beta}_E)$ for their signs. If the support $E$ were known a priori, we could determine CIs for $\beta_E$ using the asymptotic normality of $\bar{\beta}_E = (X_E^\top X_E)^{-1} X_E^\top y$. However, when $E$ is based on data, these 'classical' CIs will no longer be valid. In this case, one approach is to condition on the result of the initial model selection, i.e., knowledge of $E$ and $\hat{z}_E$. In practice, this means sampling from the so-called selective distribution, which has support $\mathcal{D} = \{(\hat{\beta}_E, \hat{z}_{-E}) : \text{sign}(\hat{\beta}_E) = \hat{z}_E , ||\hat{z}_{-E}||_\infty \leq 1\}$, and density proportional to [e.g., 90, Sec. 4.2]

$$\hat{g}(\hat{\beta}_E, \hat{z}_{-E}) \propto g\left( \varepsilon (\hat{\beta}_E, 0)^\top - X^\top (y - X_E \hat{\beta}_E) + \lambda (\hat{z}_E, \hat{z}_{-E})^\top \right). \tag{14}$$

**Synthetic Example**. We first consider the model setup described in [79, Sec. 5.3]; see App. F.2 for full details. In Fig. 2, we plot the energy distance between samples obtained using Coin MSVGD and projected Coin SVGD, and a set of 1000 samples obtained using NUTS [42], for a two-dimensional selective distribution. Similar to our previous examples, the performance of MSVGD is very sensitive to the choice of learning rate. If the learning rate is too small, the particle converge rather slowly; on the other hand, if the learning rate is too big, the particles may not even converge. Meanwhile, Coin MSVGD converges rapidly to the true target, with no need to tune a learning rate. Similar to the examples considered in Sec. 6.1, the projected methods once again fail to converge to the true target, highlighting the need for the mirrored approach.

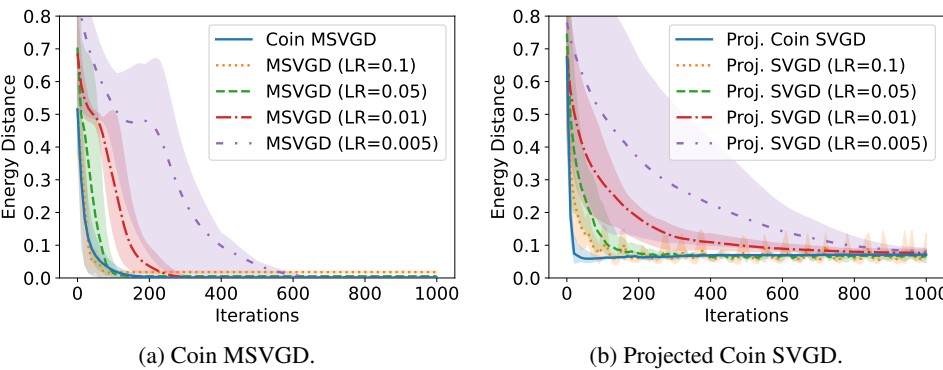

(a) Coin MSVGD.

(b) Projected Coin SVGD.

Figure 2: **Results for a two-dimensional post-selection inference target**. Energy distance versus iterations for (a) Coin MSVGD and MSVGD and (b) projected Coin SVGD and projected SVGD, for a selective distribution of the randomised Lasso in which two features are selected.

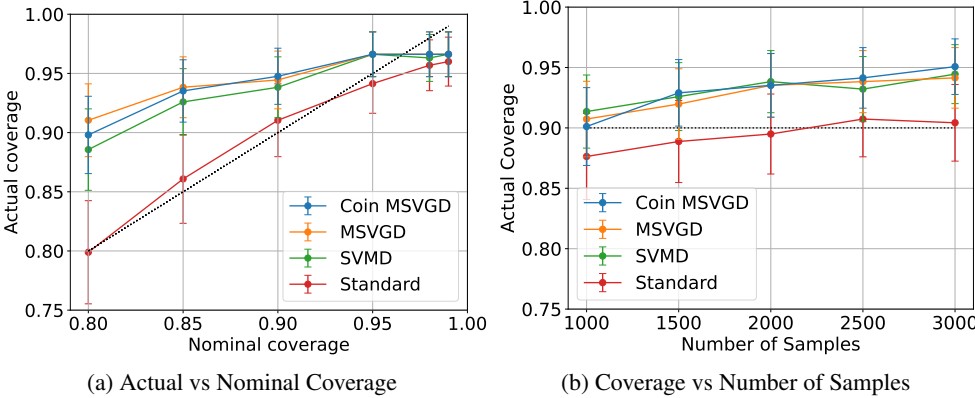

(a) Actual vs Nominal Coverage
(b) Coverage vs Number of Samples

Figure 3: **Coverage results for post-selection inference.** Coverage of the post-selection confidence intervals obtained by Coin MSVGD, MSVGD, and SVMD, for 100 repeats of the simulation in [79, Sec. 5.3].

In Fig. 3, we plot the coverage of the CIs obtained using Coin MSVGD, MSVGD, SVMD, and the `norejection` MCMC algorithm in `selectiveInference` [91], as we vary the nominal coverage or the total number of samples. For MSVGD and SVMD, we use RMSProp [92] to adapt the learning rate, with an initial learning rate of $\gamma = 0.1$, following [82]. As the nominal coverage varies, Coin MSVGD achieves similar coverage to MSVGD and SVMD; and significantly higher actual coverage than `norejection` (Fig. 3a). Meanwhile, as the number of samples varies, Coin MSVGD, MSVGD, and SVMD consistently all obtain a higher coverage than the fixed nominal coverage of 90% (Fig. 3b). This is important for small sample sizes, where the standard approach undercovers. The performance of MSVGD and SVMD is, of course, highly dependent on an appropriate choice of learning rate. In Fig. 11 (App. G.3), we provide additional plots illustrating how the coverage of the CIs obtained using MSVGD and SVMD can deteriorate when the learning-rate is chosen sub-optimally.

**Real Data Example**. We next consider a post-selection inference problem involving the HIV-1 drug resistance dataset studied in [8, 75]; see App. F.2 for full details. The goal is to identify statistically significant mutations associated with the response to the drug Lamivudine (3TC). The randomised Lasso selects a subset of mutations, for which we compute 90% CIs using 5000 samples and five methods: 'unadjusted' (the unadjusted CIs), 'standard' (the method in `selectiveInference`), MSVGD, SVMD, and Coin MSVGD. Our results (Fig. 4) indicate that the CIs obtained by Coin MSVGD are similar to those obtained via the standard approach, which we view as a benchmark, as well as by MSVGD and SVMD (with a well chosen learning rate). Importantly, they differ from the CIs obtained using the unadjusted approach (see, e.g., mutation P65R or P184V in Fig. 4). Meanwhile, for sub-optimal choices of the learning rate (Fig. 4b and Fig. 4c, Fig. 12 in App. G.3), the CIs obtained using MSVGD and SVMD differ substantially from those obtained using the baseline 'standard' approach and by Coin MSVGD, once more highlighting the sensitivity of these methods to the choice of an appropriate learning rate.

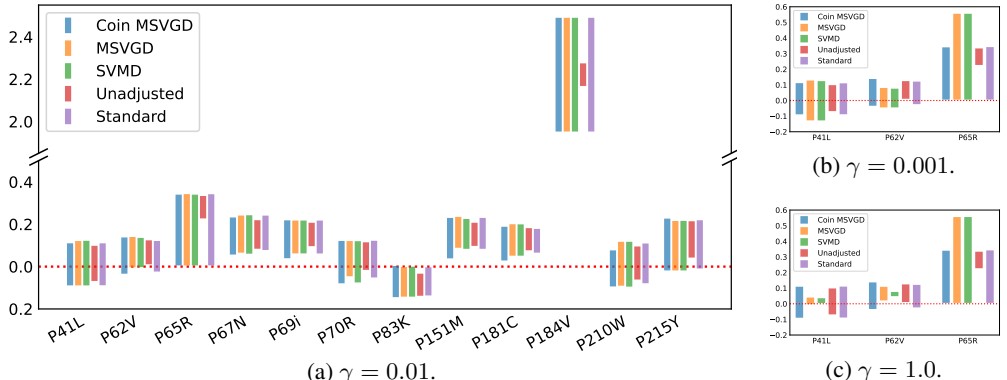

(a) $\gamma = 0.01$.

(b) $\gamma = 0.001$.

(c) $\gamma = 1.0$.

Figure 4: **Real data results for post-selection inference.** Confidence intervals for the mutations selected by the randomised Lasso as candidates for HIV-1 drug resistance. We report (a) all mutations when MSVGD and SVMD use a well chosen learning rate ($\gamma = 0.01$) and (b) - (c) a subset of mutations when MSVGD and SVMD use a smaller ($\gamma = 0.001$) or larger ($\gamma = 1.0$) learning rate.

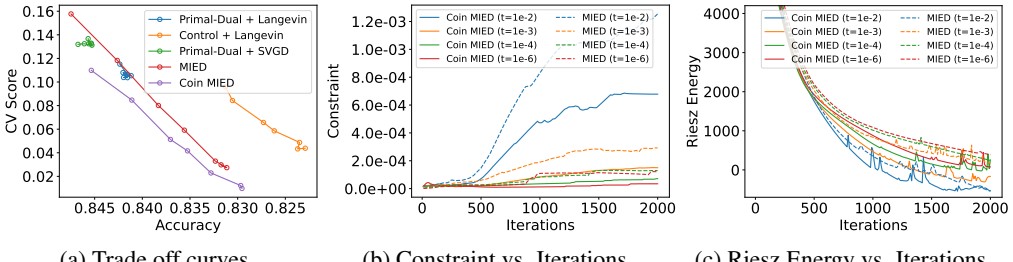

(a) Trade off curves.    (b) Constraint vs. Iterations.    (c) Riesz Energy vs. Iterations.

Figure 5: **Results for the fairness Bayesian neural network.** (a) Trade-off curves showing disparate impact versus test accuracy for MIED, Coin MIED, and the methods in [65]. (b) - (c) $\text{Cov}_{(x,y,z)\sim\mathcal{D}}[z, \hat{y}(x;\theta)]^2$ and MIE versus the number of iterations, for different values of $t$, for MIED and Coin MIED.

## 6.3 Fairness Bayesian Neural Network

Finally, following [60, 64, 65, 69], we train a fairness Bayesian neural network to predict whether an individual's income is greater than \$50,000, with gender as a protected characteristic. We use the Adult Income dataset [50]. The dataset is of the form $\mathcal{D} = \{x_i, y_i, z_i\}_{i=1}^n$, where $x_i$ denote the feature vectors, $y_i$ denote the labels (i.e., whether the income is greater than \$50,000), and $z_i$ denote the protected attribute (i.e., the gender). We train a two-layer Bayesian neural network $\hat{y}(x;w)$ with weights $w$ and place a standard Gaussian prior on each weight independently. The fairness constraint is given by $g(\theta) = \text{Cov}_{(x,y,z)\sim\mathcal{D}}[z, \hat{y}(x;\theta)]^2 - t \leq 0$ for some user-specified $t > 0$. In testing, we evaluate each method using a Calder-Verwer (CV) score [14], a standard measure of disparate impact. We run each method for $T = 2000$ iterations and use $N = 50$ particles.

In Fig. 5a, we plot the trade-off curve between test accuracy and CV score, for $t \in \{10^{-6}, 10^{-5}, 10^{-4}, 10^{-3}, 2 \times 10^{-3}, 5 \times 10^{-3}, 10^{-2}\}$. We compare the results for Coin MIED, MIED, and the methods in [65], using the default implementations. In this experiment, Coin MIED is clearly preferable to the other methods, achieving a much larger Pareto front. In Fig. 5b and Fig. 5c, we plot the constraint and the MIE versus training iterations for both coin MIED and MIED. Once again, coin MIED tends to outperform MIED, achieving both lower values of the constraint and lower values of the energy.

## 7 Discussion

**Summary**. In this paper, we introduced several new particle-based algorithms for constrained sampling which are entirely learning rate free. Our first approach was based on the coin sampling framework introduced in [81], and a perspective of constrained sampling as a mirrored optimisation problem on the space of probability measures. Based on this perspective, we also unified several existing constrained sampling algorithms, and studied their theoretical properties in continuous time. Our second approach can be viewed as the coin sampling analogue of the recently proposed MIED algorithm [60]. Empirically, our algorithms achieved comparable or superior performance to other particle-based constrained sampling algorithms, with no need to tune a learning rate.

**Limitations**. We highlight three limitations. First, like any mirrored sampling algorithm, mirrored coin sampling (e.g., Coin MSVGD), necessarily depends on the availability of a mirror map that can appropriately capture the constraints of the problem at hand. Second, Coin MSVGD has a cost of $O(N^2)$ per update. Finally, even in the population limit, establishing the convergence of mirrored coin sampling under standard conditions (e.g., strong log-concavity or a mirrored log-Sobolev inequality) remains an open problem. We leave this as an interesting direction for future work.

## Acknowledgements

LS and CN were supported by the UK Research and Innovation (UKRI) Engineering and Physical Sciences Research Council (EPSRC), grant number EP/V022636/1. CN acknowledges further support from the EPSRC, grant number EP/R01860X/1.

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

## A  Calculus in the Wasserstein Space

We recall the following from [2, Chapter 10]. Suppose $\mu \in \mathcal{P}_2(\mathbb{R}^d)$ and $\xi \in L^2(\mu)$. Let $\mathcal{F}$ be a proper and lower semi-continuous functional on $\mathcal{P}_2(\mathbb{R}^d)$. We say that $\xi \in L^2(\mu)$ belongs to the Fréchet subdifferential of $\mathcal{F}$ at $\mu$ and write $\xi \in \partial \mathcal{F}(\mu)$ if, for any $\nu \in \mathcal{P}_2(\mathbb{R}^d)$,

$$\liminf_{\nu \to \mu} \frac{\mathcal{F}(\nu) - \mathcal{F}(\mu) - \int_{\mathbb{R}^d} \langle \xi(x), \boldsymbol{t}_\mu^\nu(x) - x \rangle \mu(\mathrm{d}x)}{W_2(\nu, \mu)} \geq 0, \tag{15}$$

where $\boldsymbol{t}_\mu^\nu : \mathbb{R}^d \to \mathbb{R}^d$ denotes the optimal transport map from $\mu$ to $\nu$ [2, Chapter 7.1]. Under mild conditions [2, Lemma 10.4.1], the subdifferential $\partial \mathcal{F}$ is single-valued, $\partial \mathcal{F}(\mu) = \{\nabla_{W_2} \mathcal{F}(\mu)\}$, and $\nabla_{W_2} \mathcal{F}(\mu) \in L^2(\mu)$ is given by

$$\nabla_{W_2} \mathcal{F}(\mu) = \nabla \frac{\partial \mathcal{F}(\mu)}{\partial \mu}(x) \quad \text{for } \mu\text{-a.e. } x \in \mathbb{R}^d, \tag{16}$$

where $\frac{\partial \mathcal{F}(\mu)}{\partial \mu} : \mathbb{R}^d \to \mathbb{R}$ denotes the first variation of $\mathcal{F}$ at $\mu$, that is, the unique function such that

$$\lim_{\varepsilon \to 0} \frac{1}{\varepsilon} \left( \mathcal{F}(\mu + \varepsilon \zeta) - \mathcal{F}(\mu) \right) = \int_{\mathbb{R}^d} \frac{\partial \mathcal{F}(\mu)}{\partial \mu}(x) \zeta(\mathrm{d}x), \tag{17}$$

where $\zeta = \nu - \mu$, and $\nu \in \mathcal{P}_2(\mathbb{R}^d)$. We will refer to $\nabla_{W_2} \mathcal{F}(\mu)$ as the Wasserstein gradient of $\mathcal{F}$ at $\mu$.

## B  Examples of Mirrored Wasserstein Gradient Flows

In this section, we outline several algorithms which arise as special cases of the MWGF introduced in Sec 3.2, namely

$$\frac{\partial \eta_t}{\partial t} + \nabla \cdot (w_t \eta_t) = 0 \quad , \quad \mu_t = (\nabla \phi^*)_\# \eta_t, \tag{18}$$

where $(w_t)_{t \geq 0}$ are vector fields chosen to ensure the convergence of $(\eta_t)_{t \geq 0}$ to $\nu = (\nabla \phi)_\# \pi$.

We will assume, throughout this section, that $\pi \propto e^{-V}$ and that $\nu = (\nabla \phi)_\# \pi \propto e^{-W}$. The Monge-Ampere equation determines the relationship between $W$ and $V$. In particular, we have [e.g., 43]

$$e^{-V} = e^{-W \circ \nabla \phi} \det \nabla^2 \phi \quad , \quad e^{-W} = e^{-V \circ \nabla \phi^*} \det \nabla^2 \phi^*. \tag{19}$$

Thus, the potential $W(y)$ of the mirrored target $\nu$ evaluated at $y = \nabla \phi(x)$ can be expressed in terms of the potential $V(x)$ of the original target $\pi$ evaluated at $x$, as

$$W(y) = V(x) + \log \det \nabla^2 \phi(x). \tag{20}$$

### B.1  Mirrored Langevin Dynamics

Suppose that $\mathcal{F}(\eta) = \mathrm{KL}(\eta \| \nu)$, where, as elsewhere, $\nu = (\nabla \phi)_\# \pi$ is the dual target. In addition, let $w_t = -\nabla_{W_2} \mathcal{F}(\eta_t)$. In this case, it is straightforward to show that [e.g. 2, Chapter 10] $\nabla_{W_2} \mathcal{F}(\eta) = \nabla \log \left( \frac{\eta}{\nu} \right)$. Thus, for this choice of $(w_t)_{t \geq 0}$, the MWGF in (18) reads

$$\frac{\partial \eta_t}{\partial t} - \nabla \cdot \left( \nabla \log \left( \frac{\eta_t}{\nu} \right) \eta_t \right) = 0, \quad \mu_t = (\nabla \phi^*)_\# \eta_t. \tag{21}$$

Substituting $\nabla \log \nu = -\nabla W$, and using the fact that $\eta_t \nabla \log(\eta_t) = \nabla \eta_t$, this can also be written as

$$\frac{\partial \eta_t}{\partial t} - \nabla \cdot (\eta_t \nabla W + \nabla \eta_t) = 0, \quad \mu_t = (\nabla \phi^*)_\# \eta_t. \tag{22}$$

This PDE is nothing more than the Fokker-Planck equation describing the evolution of the law of the overdamped Langevin SDE with respect to the mirrored target $\nu \propto e^{-W}$. In particular, suppose that $x_0 \sim \mu_0$, $y_0 = \nabla \phi(x_0) \sim \eta_0$, and that $(x_t)_{t \geq 0}$ and $(y_t)_{t \geq 0}$ are the solutions of

$$\mathrm{d}y_t = -\nabla W(y_t)\mathrm{d}t + \sqrt{2}\mathrm{d}b_t \quad , \quad x_t = \nabla \phi^*(y_t), \tag{23}$$

where $b = (b_t)_{t \geq 0}$ is a standard Brownian motion. Then, if we let $(\mu_t)_{t \geq 0}$ and $(\eta_t)_{t \geq 0}$ denote the laws of $(x_t)_{t \geq 0}$ and $(y_t)_{t \geq 0}$, it follows that $(\mu_t)_{t \geq 0}$ and $(\eta_t)_{t \geq 0}$ satisfy (22). This is precisely the mirrored Langevin dynamics (MLD) introduced in [43]. We remark that, using (20), the MLD in (23) can also be written as

$$\mathrm{d}y_t = -\nabla^2 \phi(x_t)^{-1} (\nabla V(x_t) + \nabla \log \det \nabla^2 \phi(x_t))\mathrm{d}t + \sqrt{2}\mathrm{d}b_t, \quad x_t = \nabla \phi^*(y_t). \tag{24}$$

**Remark 1.** *It is worth distinguishing between the mirrored Langevin dynamics in* (23) *and the so-called mirror Langevin diffusion [e.g.,* 44, *Equation 2], which refers to the solution of*

$$\mathrm{d}y_t = -\nabla V(x_t)\mathrm{d}t + \sqrt{2}\left[\nabla^2\phi(x_t)\right]^{\frac{1}{2}}\mathrm{d}b_t, \quad x_t = \nabla\phi^*(y_t) \tag{25}$$

*These dynamics, and the corresponding Euler-Maruyama discretisation, known as the mirror Langevin algorithm or mirror Langevin Monte Carlo, were first studied in [103], and have since also been analysed in [1, 44, 61]. An alternative time discretisation has also been studied in [19].*

*In this case, one can show that the law* $(\mu_t)_{t\geq 0}$ *of the solution of the mirror Langevin diffusion in* (25) *satisfies the Fokker-Planck equation [e.g.,* 44, *App. A.2]*

$$\frac{\partial\mu_t}{\partial t} + \nabla\cdot(\mu_t v_t) = 0, \quad v_t = -[\nabla^2\phi]^{-1}\nabla\log\frac{\mu_t}{\pi}. \tag{26}$$

*Recalling that* $\nabla_{W_2}\mathrm{KL}(\mu_t||\pi) = \nabla\log\frac{\mu_t}{\pi}$, *one can view* (26) *as a special case of the so-called Wasserstein mirror flow (WMF), defined according to [e.g.* 1, *App. C]*

$$\frac{\partial\mu_t}{\partial t} + \nabla\cdot(v_t\mu_t) = 0, \quad v_t = -(\nabla^2\phi)^{-1}\nabla_{W_2}\mathcal{F}(\mu_t). \tag{27}$$

*The WMF is rather different from the MWGF in* (18). *In particular, the WMF in* (27) *describes the evolution of* $\mu_t$ *according to the Wasserstein tangent vector* $-(\nabla^2\phi)^{-1}\nabla_{W_2}\mathcal{F}(\mu_t)$, *where* $\mathcal{F}$ *is a functional minimised at* $\pi$, *while the MWGF in* (18) *describes the evolution of the mirrored* $\eta_t := (\nabla\phi)_{\#}\mu_t$ *according to the tangent vector* $-\nabla_{W_2}\mathcal{F}(\eta_t)$, *where* $\mathcal{F}$ *is now a functional minimised at* $\nu = (\nabla\phi)_{\#}\pi$. *We refer to [19] for a detailed discussion of the mirror Langevin diffusion from this perspective; and [26] for a more general formulation of the WMF.*

### B.1.1 Continuous Time Results

We now study the properties of the dynamics in (23) in continuous time, starting with the dissipation of $\mathrm{KL}(\mu_t|\pi)$ and $\chi^2(\mu_t|\pi)$ along the MLD. These results are a natural extension of existing results in the unconstrained case to our setting.

**Proposition 1.** *The dissipation of* $\mathrm{KL}(\cdot||\pi)$ *along the MWGF in* (21) *is given by*

$$\frac{\mathrm{dKL}(\mu_t||\pi)}{\mathrm{d}t} = -\left|\left|[\nabla^2\phi]^{-1}\nabla\log\left(\frac{\mu_t}{\pi}\right)\right|\right|^2_{L^2(\mu_t)}. \tag{28}$$

*Proof.* Using differential calculus in the Wasserstein space and the chain rule, we have that

$$\frac{\mathrm{dKL}(\eta_t||\nu)}{\mathrm{d}t} = \left\langle w_t, \nabla\log\left(\frac{\eta_t}{\nu}\right)\right\rangle_{L^2(\eta_t)} = -\left|\left|\nabla\log\left(\frac{\eta_t}{\nu}\right)\right|\right|^2_{L^2(\eta_t)}. \tag{29}$$

By [43, Theorem 2], we have that $\mathrm{KL}(\eta||\nu) = \mathrm{KL}(\mu||\pi)$. To deal with the term on the RHS, first note that, using the formula for the change of variable in a probability density, we have that

$$\eta(y) = \mu(\nabla\phi^*(y))|\det\nabla^2\phi(\nabla\phi^*(y))|^{-1} \quad , \quad \nu(y) = \pi(\nabla\phi^*(y))|\det\nabla^2\phi(\nabla\phi^*(y))|^{-1}. \tag{30}$$

Thus, in particular, if $y = \nabla\phi(x)$, then we have that

$$\log\left(\frac{\eta}{\nu}\right)(y) = \log\left(\frac{\mu}{\pi}\right)(\nabla\phi^*(y)) = \log\left(\frac{\mu}{\pi}\right)(x). \tag{31}$$

Using this, and the fact that $\eta = (\nabla\phi)_{\#}\mu$, we can now compute

$$\left|\left|\nabla\log\left(\frac{\eta_t}{\nu}\right)\right|\right|^2_{L^2(\eta_t)} = \int\left|\left|\nabla_y\log\left(\frac{\eta_t}{\nu}\right)(y)\right|\right|^2\mathrm{d}\eta_t(y) \tag{32}$$

$$= \int\left|\left|\nabla_y\log\left(\frac{\eta_t}{\nu}\right)(y)\right|\right|^2\mathrm{d}((\nabla\phi)_{\#}\mu_t)(y) \tag{33}$$

$$= \int\left|\left|\nabla_y\log\left(\frac{\eta_t}{\nu}\right)(\nabla\phi(x))\right|\right|^2\mathrm{d}\mu_t(x) \tag{34}$$

$$= \int\left|\left|\nabla_y\log\left(\frac{\mu_t}{\pi}\right)(x)\right|\right|^2\mathrm{d}\mu_t(x) \tag{35}$$

$$= \int\left|\left|[\nabla^2\phi(x)]^{-1}\nabla_x\log\left(\frac{\mu_t}{\pi}\right)(y)\right|\right|^2\mathrm{d}\mu_t(y) \tag{36}$$

$$= \left|\left|[\nabla^2\phi(y)]^{-1}\nabla_y\log\left(\frac{\mu_t}{\pi}\right)(y)\right|\right|^2_{L^2(\mu_t)}. \tag{37}$$

The conclusion now follows. □

**Proposition 2.** *The dissipation of $\chi^2(\cdot||\pi)$ along the MWGF in* (21) *is given by*

$$\frac{\mathrm{d}\chi^2(\mu_t||\pi)}{\mathrm{d}t} = -2\left|\left|[\nabla^2\phi]^{-1}\nabla\frac{\mu_t}{\pi}\right|\right|^2_{L^2(\pi)}. \tag{38}$$

*Proof.* The proof is rather similar to the previous one. In this case, using the fact that $\nabla_{W_2}\mathcal{X}^2(\eta||\nu) = 2\nabla\frac{\eta}{\nu}$, we have

$$\frac{\mathrm{d}\chi^2(\eta_t||\nu)}{\mathrm{d}t} = \left\langle -\nabla\log\left(\frac{\eta_t}{\nu}\right), 2\nabla\frac{\eta_t}{\nu}\right\rangle_{L^2(\eta_t)} = -2\left|\left|\nabla\frac{\eta_t}{\nu}\right|\right|^2_{L^2(\nu)}.$$

We next observe, using now the fact that $\frac{\eta}{\nu}(y) = \frac{\mu}{\pi}(x)$ which follows from (30), that

$$\chi^2(\eta_t||\nu) = \int\left(\frac{\eta_t}{\nu}(y) - 1\right)^2 \mathrm{d}\nu(y) = \int\left(\frac{\eta_t}{\nu}(y) - 1\right)^2 \mathrm{d}((\nabla\phi)_{\#}\pi)(y) \tag{39}$$

$$= \int\left(\frac{\eta_t}{\nu}(\nabla\phi(x)) - 1\right)^2 \mathrm{d}\pi(x) = \int\left(\frac{\mu_t}{\pi}(x) - 1\right)^2 \mathrm{d}\pi(x) = \chi^2(\mu_t||\pi). \tag{40}$$

In addition, based on a very similar argument, we have that

$$\left|\left|\nabla\left(\frac{\eta_t}{\nu}\right)\right|\right|^2_{L^2(\nu)} = \int\left|\left|\nabla_y\left(\frac{\eta_t}{\nu}\right)(y)\right|\right|^2 \mathrm{d}\nu(y) = \int\left|\left|\nabla_y\left(\frac{\eta_t}{\nu}\right)(y)\right|\right|^2 \mathrm{d}((\nabla\phi)_{\#}\pi)(y) \tag{41}$$

$$= \int\left|\left|\nabla_y\left(\frac{\eta_t}{\nu}\right)(\nabla\phi(x))\right|\right|^2 \mathrm{d}\pi(x) = \int\left|\left|\nabla_y\left(\frac{\mu_t}{\pi}\right)(x)\right|\right|^2 \mathrm{d}\pi(x) \tag{42}$$

$$= \int\left|\left|[\nabla^2\phi(y)]^{-1}\nabla_y\left(\frac{\mu_t}{\pi}\right)(y)\right|\right|^2 \mathrm{d}\pi(y) = \left|\left|[\nabla^2\phi(y)]^{-1}\nabla_y\left(\frac{\mu_t}{\pi}\right)(y)\right|\right|^2_{L^2(\pi)}. \tag{43}$$

This completes the proof. □

Since the RHS of both (28) and (38) are non-positive, these results show that $\mathrm{KL}(\mu_t|\pi)$ and $\chi^2(\mu_t|\pi)$ decrease along the MLD. As an immediate corollary, we have the following continuous time convergence rate for the time-average of $||[\nabla^2\phi]^{-1}\nabla\log(\frac{\mu_t}{\pi})||^2_{L^2(\mu_t)}$ and $||[\nabla^2\phi]^{-1}\nabla(\frac{\mu_t}{\pi})||^2_{L^2(\pi)}$.

**Proposition 3.** *For any $t \geq 0$, it holds that*

$$\min_{0\leq s\leq t}\left|\left|[\nabla^2\phi]^{-1}\nabla\log\left(\frac{\mu_t}{\pi}\right)\right|\right|^2_{L^2(\mu_t)} \leq \frac{1}{t}\int_0^t\left|\left|[\nabla^2\phi]^{-1}\nabla\log\left(\frac{\mu_s}{\pi}\right)\right|\right|^2_{L^2(\mu_s)} \mathrm{d}s \leq \frac{\mathrm{KL}(\mu_0||\pi)}{t}. \tag{44}$$

**Proposition 4.** *For any $t \geq 0$, it holds that*

$$\min_{0\leq s\leq t}\left|\left|[\nabla^2\phi]^{-1}\nabla\left(\frac{\mu_t}{\pi}\right)\right|\right|^2_{L^2(\pi)} \leq \frac{1}{t}\int_0^t\left|\left|[\nabla^2\phi]^{-1}\nabla\left(\frac{\mu_s}{\pi}\right)\right|\right|^2_{L^2(\pi)} \mathrm{d}s \leq \frac{\chi^2(\mu_0||\pi)}{t}. \tag{45}$$

*Proof.* These results follow straightforwardly from integration of (28) or (38) and the non-negativity of the KL or $\chi^2$ divergence. □

To obtain stronger convergence results, we will require additional assumptions on the target measure. In the first case, we can assume that the target satisfies what we will refer to as a *mirrored logarithmic Sobolev inequality* (LSI).

**Proposition 5.** *Assume that $\pi$ satisfies a mirrored LSI with constant $\lambda > 0$. In particular, for all $\mu \in \mathcal{P}_2(\mathcal{X})$, there exists $\lambda > 0$ such that*

$$\mathrm{KL}(\mu||\pi) \leq \frac{1}{2\lambda}\left|\left|[\nabla^2\phi]^{-1}\nabla\log\left(\frac{\mu}{\pi}\right)\right|\right|^2_{L^2(\mu)}. \tag{46}$$

*Then the KL divergence converges exponentially along the MWGF in* (21)*:*

$$\mathrm{KL}(\mu_t||\pi) \leq e^{-2\lambda t}\mathrm{KL}(\mu_0||\pi). \tag{47}$$

*Proof.* Substituting the mirrored LSI (46) into (28), we have that

$$\frac{\mathrm{d}\mathrm{KL}(\mu_t||\pi)}{\mathrm{d}t} = - \left|\left|[\nabla^2\phi]^{-1}\nabla\log\left(\frac{\mu_t}{\pi}\right)\right|\right|^2_{L^2(\mu_t)} \leq -2\lambda\mathrm{KL}(\mu_t||\pi). \tag{48}$$

The conclusion now follows straightforwardly from Grönwall's inequality [39]. $\square$

Alternatively, under the assumption that the target satisfies a *mirrored Poincaré inequality* (PI), we can obtain exponential convergence in a number of metrics.

**Proposition 6.** *Assume that $\pi$ satisfies a mirrored PI with constant $\kappa > 0$. In particular, for all locally Lipschitz $g \in L^2(\pi)$, there exists $\kappa > 0$ such that*

$$\mathrm{Var}_\pi[g] \leq \frac{1}{\kappa}\left|\left|[\nabla^2\phi]^{-1}\nabla g\right|\right|^2_{L^2(\pi)} \tag{49}$$

*Then the total variation distance, Hellinger distance, KL divergence, and the $\chi^2$ divergence all converge exponentially along the MWGF in (21):*

$$\max(2||\mu_t - \pi||^2_{\mathrm{TV}}, \ H^2(\mu_t, \pi), \ \frac{\lambda}{2}W_2^2(\mu_t, \pi), \ \mathrm{KL}(\mu_t||\pi), \ \chi^2(\mu_t||\pi)) \leq e^{-2\kappa t}\chi^2(\mu_0||\pi). \tag{50}$$

*Proof.* The proof proceeds similarly to above. In particular, this time substituting the mirrored PI (49) with $g = \frac{\mu_t}{\pi} - 1$ into (38), we have

$$\frac{\mathrm{d}\chi^2(\mu_t||\pi)}{\mathrm{d}t} = -2\left|\left|[\nabla^2\phi]^{-1}\nabla\frac{\mu_t}{\pi}\right|\right|^2_{L^2(\pi)} \leq -2\kappa\chi^2(\mu_t||\pi).$$

The result for the $\chi^2$ divergence follows via an application of Grönwall's inequality. The remaining bounds follow from standard comparison inequalities [e.g., 93, Sec. 2.4]. $\square$

**Remark 2.** *The assumption that $\pi$ satisfies the mirrored LSI in (46) or the mirrored PI in (49) is precisely equivalent to the assumption that the mirrored target $\nu = (\nabla\phi)_\#\pi$ satisfies the classical LSI or the classical PI, respectively. The LSI holds, for example, for all strongly log-concave distributions, with constant equal to the reciprocal of the strong convexity constant of the potential [4]. The class of measures satisfying the PI is even larger, including all strongly log-concave measures and, more generally, all log-concave measures [10, 17, 47].*

**Remark 3.** *The mirrored LSI in (46) and the mirrored PI in (49) are distinct from the so-called mirror LSI [44, Assumption 2] and mirror PI [19, Definition 1]. These assumptions are used in [44, Proposition 1] and [19, Theorem 1], respectively, to establish exponential convergence of the mirror Langevin diffusion [e.g., 44, Equation 2], which we recall is distinct from the* mirrored *Langevin diffusion in (23); see Remark 1.*

**Remark 4.** *In Propositions 5 - 6 above, and also in Proposition 7 below, our assumptions on the target $\pi$ are, in some sense, better viewed as assumptions on the mirrored target $\nu = (\nabla\phi)_\#\pi$. Indeed, by construction, the imposed assumptions are equivalent to the assumptions that the mirrored target $\nu$ satisfies a LSI (Proposition 5, 7), or that the mirrored target $\nu$ satisfies a PI (Proposition 6).*

*In some sense, it would be preferable to establish such results under more direct assumptions on the target $\pi$ itself. One possibility is to assume that the target $\pi$ is strongly log-concave. In this case, [43, Proof of Theorem 3] guarantees the existence of a mirror map such that the mirrored target $\nu = (\nabla\phi)_\#\pi$ is also strongly log-concave. Thus, in particular, the mirrored target satisfies a LSI and a PI or, equivalently, the target $\pi$ satisfies a mirrored LSI and a mirrored PI. Thus, Propositions 5 - 7 all still hold.*

### B.1.2 Discrete Time Results

In practice, of course, it is necessary to discretise the dynamics in (23) in time. Applying a standard Euler-Maruyama discretisation to (23), or equivalently a forward-flow discretisation to (21) [e.g., 99], one arrives at the *mirrored Langevin algorithm* (MLA) from [43], viz

$$y_{t+1} = y_t - h\nabla W(y_t) + \sqrt{2h}\xi_t, \quad x_{t+1} = \nabla\phi^*(y_{t+1}) \tag{51}$$

where $h > 0$ is the step size or learning rate and where $(\xi_t)_{t\in\mathbb{N}}$ are a sequence of i.i.d. standard normal random variables.

**Proposition 7.** *Suppose that $\pi$ satisfies the mirrored LSI with constant $\lambda > 0$. In addition, suppose that $\pi$ is mirror-$L$-smooth.[2] Then, if $0 < \gamma \leq \frac{\lambda}{4L^2}$, the iterates $(x_t)_{t \geq 0}$ in (51) satisfy*

$$\mathrm{KL}(\mu_t || \pi) \leq e^{-\lambda \gamma t} \mathrm{KL}(\mu_0 || \pi) + \frac{8\gamma d L^2}{\lambda}. \tag{52}$$

*Proof.* By [43, Theorem 2], and (32) - (37) in the proof of Proposition 1, if $\pi$ satisfies a mirrored LSI with constant $\lambda > 0$, then $\nu = (\nabla \phi)_{\#} \pi$ satisfies a LSI with the same constant $\lambda$. In addition, by definition, if $\pi$ is mirror-$L$-smooth, then $\nu$ is $L$-smooth. Thus, it follows from [94, Theorem 1] that

$$\mathrm{KL}(\eta_t || \nu) \leq e^{-\lambda \gamma t} \mathrm{KL}(\eta_0 || \nu) + \frac{8\gamma d L^2}{\lambda}. \tag{53}$$

Finally, using the invariance of the KL divergence under the mirror map $\nabla \phi$ [43, Theorem 2], the result follows. □

**Remark 5.** *Using a similar approach, one can also obtain bounds on $\chi^2(\mu_t || \pi)$ under a mirrored LSI [35, Theorem 4], and for the Renyi divergence $\mathcal{R}_q(\mu_t || \pi)$ under a mirrored LSI [94, Theorem 2] or a mirrored PI [94, Theorem 3].*

**Remark 6.** *One could also consider a forward Euler discretisation of the dynamics in (21), which leads to a mirrored version of the so-called deterministic Langevin Monte Carlo algorithm [40].*

## B.2 Mirrored Stein Variational Gradient Descent

We now turn our attention to the MSVGD dynamics given in Sec. 3.2.1, which we recall are defined by the continuity equation

$$\frac{\partial \eta_t}{\partial t} - \nabla \cdot \left( P_{\eta_t, k_\phi} \nabla \log \left( \frac{\eta_t}{\nu} \right) \eta_t \right) = 0, \quad \mu_t = (\nabla \phi^*)_{\#} \eta_t, \tag{54}$$

where, as previously, $k_\phi$ is the kernel defined according to $k_\phi(y, y') = k(\nabla \phi^*(y), \nabla \phi^*(y'))$, given some base kernel $k : \mathcal{X} \times \mathcal{X} \to \mathbb{R}$.

### B.2.1 Continuous Time Results

We now study the convergence properties of the continuous-time MSVGD dynamics. We note that a similar set of results can also found in [82, App. H]. We provide them here to highlight the analogues with the convergence results obtained for the MLD in Sec. B.1.

**Proposition 8.** *The dissipation of the KL along the mirrored SVGD gradient flow is given by*

$$\frac{\mathrm{dKL}(\mu_t || \pi)}{\mathrm{d}t} = - \left\| S_{\mu_t, k} [\nabla^2 \phi]^{-1} \nabla \log \left( \frac{\mu_t}{\pi} \right) \right\|_{\mathcal{H}_k^d}^2. \tag{55}$$

*Proof.* The proof is similar to the proof of Proposition 1. In particular, using differential calculus on the Wasserstein space and the chain rule, we have

$$\frac{\mathrm{dKL}(\eta_t || \nu)}{\mathrm{d}t} = \left\langle w_t, \nabla \log \left( \frac{\eta_t}{\nu} \right) \right\rangle_{L^2(\eta_t)} = - \left\| S_{\eta_t, k_\phi} \nabla \log \left( \frac{\eta_t}{\nu} \right) \right\|_{\mathcal{H}_{k_\phi}^d}^2. \tag{56}$$

By Theorem 2 in [43], we have that $\mathrm{KL}(\eta || \nu) = \mathrm{KL}(\mu || \pi)$. In addition, using (31), we can compute

$$\left\| S_{\eta_t, k_\phi} \nabla \log \left( \frac{\eta_t}{\nu} \right) \right\|_{\mathcal{H}_{k_\phi}^d}^2 \tag{57}$$

$$= \int \int k(\nabla \phi^*(y), \nabla \phi^*(y')) \left\langle \nabla_y \log \left( \frac{\eta_t}{\nu} \right)(y), \nabla_{y'} \log \left( \frac{\eta_t}{\nu} \right)(y') \right\rangle \mathrm{d}\eta_t(y) \mathrm{d}\eta_t(y') \tag{58}$$

$$= \int \int k(x, x') \left\langle \nabla_y \log \left( \frac{\mu_t}{\pi} \right)(x), \nabla_{y'} \log \left( \frac{\mu_t}{\pi} \right)(x') \right\rangle \mathrm{d}\mu_t(x) \mathrm{d}\mu_t(x') \tag{59}$$

$$= \int \int k(x, x') \left\langle [\nabla^2 \phi(x)]^{-1} \nabla_x \log \left( \frac{\mu_t}{\pi} \right)(x), [\nabla^2 \phi(x')]^{-1} \nabla_{x'} \log \left( \frac{\mu_t}{\pi} \right)(x') \right\rangle \mathrm{d}\mu_t(x) \mathrm{d}\mu_t(x')$$

$$= \left\| S_{\mu_t, k} [\nabla^2 \phi]^{-1} \nabla \log \left( \frac{\mu_t}{\pi} \right) \right\|_{\mathcal{H}_k^d}^2. \tag{60}$$

---

[2] We say that $\pi \propto e^{-V}$ is mirror-$L$-smooth if $\nu \propto e^{-W}$ is $L$-smooth. This holds, in particular, if $W = -V \circ \nabla \phi^* + \log \det \nabla^2 \phi^*$ has Hessian bounded by $L$.

$\square$

**Remark 7.** *The quantity on the RHS of* (55) *is referred to in [86] as the* mirrored Stein Fisher information *of $\mu$ with respect to $\pi$, with mirror map $\nabla\phi$. This quantity represents the Stein Fisher information [32] between $(\nabla\phi)_{\#}\mu$ and $(\nabla\phi)_{\#}\pi$, given the kernel $k_\phi(\cdot,\cdot) = k(\nabla\phi^*(\cdot), \nabla\phi^*(\cdot))$.*

Similar to before, a straightforward consequence of this result is a continuous time convergence rate for the average of the mirrored stein discrepancy along the continuous-time MSVGD dynamics.

**Proposition 9.** *For all $t \geq 0$, it holds that*

$$\min_{0 \leq s \leq t} \left\| S_{\mu_s,k}[\nabla^2\phi]^{-1}\nabla \log\left(\frac{\mu_s}{\pi}\right) \right\|_{\mathcal{H}_k^d}^2 \leq \frac{1}{t}\int_0^t \left\| S_{\mu_s,k}[\nabla^2\phi]^{-1}\nabla\log\left(\frac{\mu_s}{\pi}\right) \right\|_{\mathcal{H}_k^d}^2 \mathrm{d}s \leq \frac{\mathrm{KL}(\mu_0||\pi)}{t}. \tag{61}$$

*Proof.* We integrate (55), and use the non-negativity of the KL divergence. $\square$

To obtain stronger convergence guarantees, we must once again assume some additional properties on the target. Here, a natural choice is the *mirrored Stein LSI*, the analogue of the Stein LSI introduced in [32] in the mirrored setting; see also [82, Definition 2].

**Proposition 10.** *Assume that $\pi$ satisfies the mirrored Stein LSI with constant $\lambda > 0$. In particular, for all $\mu \in \mathcal{P}_2(\mathcal{X})$, there exists $\lambda > 0$ such that*

$$\mathrm{KL}(\mu||\pi) \leq \frac{1}{2\lambda}\left\| S_{\mu,k}[\nabla^2\phi]^{-1}\nabla\log\left(\frac{\mu}{\pi}\right) \right\|_{\mathcal{H}_k^d}^2. \tag{62}$$

*Then the KL divergence converges exponentially along the continuous-time MSVGD dynamics:*

$$\mathrm{KL}(\mu_t||\pi) \leq e^{-2\lambda t}\mathrm{KL}(\mu_0||\pi). \tag{63}$$

*Proof.* The result is a straightforward consequence of (55) and (62). In particular, substituting (62) into (55), we have

$$\frac{\mathrm{dKL}(\mu_t||\pi)}{\mathrm{d}t} = -\left\| S_{\mu_t,k}[\nabla^2\phi]^{-1}\nabla\log\left(\frac{\mu_t}{\pi}\right) \right\|_{\mathcal{H}_k^d}^2 \leq 2\lambda\mathrm{KL}(\mu_t||\pi). \tag{64}$$

We conclude by applying Grönwall's inequality. $\square$

**Remark 8.** *The assumption that $\pi$ satisfies the mirrored Stein LSI is equivalent to the assumption that $\nu = (\nabla\phi)_{\#}\pi$ satisfies the classical Stein LSI with kernel $k_\phi$. The Stein LSI is rather less well known than the classical LSI, and holds for a much smaller class of targets. In particular, this condition fails to hold if the kernel $k$ is too regular with respect to the target $\pi$, e.g., if $\pi$ has exponential tails and the derivatives of the kernel $k$ and the potential $V$ grow at most polynomially [32, 52]. It remains an open problem to establish conditions for which this conditions holds.*

### B.2.2 Discrete Time Results

In discrete time, [86] recently established a descent lemma and a complexity bound for MSVGD in the population limit. For completeness, we recall one of these results here.

**Theorem 1** (Corollary 1 in [86]). *Suppose that the following assumptions hold:*

*(1) The mirror function $\phi: \mathcal{X} \to (-\infty, +\infty)$ is strongly $K$-convex.*

*(2) There exist $B_1, B_2 > 0$ such that for all $x \in \mathcal{X}$, $k(x,x') \leq B_1^2$ and $\partial_{x_i}\partial_{x_j}k(x,x')|_{x=x'} \leq B_2^2$ for all $i \in [d]$, $j \in [d]$.*

*(3) There exist $L_0, L_1 \geq 0$ such that $||\nabla^2 W(y)|| \leq L_0 + L_1||\nabla W(y)||$.*

*(4) There exists $p \geq 1$, $y_0 \in \mathbb{R}^d$, and $s > 0$ such that $\int \exp\left[s||y - y_0||^p\right]\mathrm{d}\nu(y) < \infty$.*

*(5) There exists $p \geq 1$, $C_p > 0$ such that $||\nabla W(y)|| \leq C_p\left(||y||^p + 1\right)$ for any $y \in \mathbb{R}^d$.*

*Then, provided $\gamma$ satisfies [86, Equation 21], then*

$$\frac{1}{T}\sum_{t=0}^{T-1}\left|\left|S_{\mu_t,k}[\nabla^2\phi]^{-1}\nabla\log\left(\frac{\mu_t}{\pi}\right)\right|\right|_{\mathcal{H}_k^d}^2 \leq \frac{2\mathrm{KL}(\mu_0||\pi)}{T\gamma}. \tag{65}$$

**Remark 9.** *The results in [86] are obtained by extending the proofs used to establish convergence of SVGD in [85] to the mirrored setting. In this case, essentially all of the same arguments can be applied (in the dual space), provided suitable assumptions are imposed on the mirrored target $\nu = (\nabla\phi)_{\#}\pi$. The disadvantage of this approach is that the required assumptions are specified in terms of the mirrored target $\nu$, rather than the target $\pi$.*

*Following a similar approach to [43, Theorem 3], let us show how to recover this result under a standard assumption on $\pi$ itself. Suppose we replace [86, Assumption 4] by the assumption that $\pi$ is strongly log-concave. Under this assumption, [43, Proof of Theorem 3] guarantees the existence of a mirror map such that $\nu$ is also strongly log-concave. By the Bakry-Emery criterion [4], it follows that the $T_2$ inequality is satisfied by the dual target $\nu$, and thus so too is the $T_1$ inequality. Finally, the $T_1$ inequality is necessary and sufficient for [86, Assumption 4] by [95, Theorem 22.10]. Thus, [86, Corollary 1] holds whenever the target $\pi$ is strongly log-concave.*

### B.3 Mirrored Laplacian Adjusted Wasserstein Gradient Descent

We now consider the MWGF of the chi-squared divergence; see also [20, Sec. 3] or [2, Theorem 11.2.1] in the unconstrained case. Following similar steps to [20], we will then obtained a mirrored version of the Laplacian Adjusted Wasserstein Gradient Descent (LAWGD) algorithm.

### B.3.1 MWGF of the Chi-Squared Divergence

Suppose that $\mathcal{F}(\eta) = \chi^2(\eta||\nu)$ with $\nabla_{W_2}\mathcal{F}(\eta) = 2\nabla\left(\frac{\eta}{\nu}\right)$. Setting $w_t = -\nabla_{W_2}\mathcal{F}(\eta_t)$ in (4), we are interested in the following MWGF

$$\frac{\partial\eta_t}{\partial t} - 2\nabla\cdot\left(\nabla\left(\frac{\eta_t}{\nu}\right)\eta_t\right) = 0, \quad \mu_t = (\nabla\phi)_{\#}\eta_t. \tag{66}$$

Similar to before, we will begin by studying the convergence properties of this MWGF in continuous time.

**Proposition 11.** *The dissipation of $\mathrm{KL}(\cdot||\pi)$ along the MWGF in (66) is given by*

$$\frac{d\mathrm{KL}(\mu_t||\pi)}{dt} = -2\left|\left|[\nabla^2\phi]^{-1}\nabla\frac{\mu_t}{\pi}\right|\right|_{L^2(\pi)}^2. \tag{67}$$

*Proof.* The proof is essentially identical to the proof of Proposition 2, although the roles of the Lyapunov functional and the Wasserstein gradient are now reversed. In particular, we now have

$$\frac{d\mathrm{KL}(\eta_t||\nu)}{dt} = \left\langle 2\nabla\frac{\eta_t}{\nu}, -\nabla\log\left(\frac{\eta_t}{\nu}\right)\right\rangle_{L^2(\eta_t)} = -2\left|\left|\nabla\frac{\eta_t}{\nu}\right|\right|_{L^2(\nu)}^2.$$

By Theorem 2 in [43] and (41) - (43) in the proof of Proposition 2, we have that $\mathrm{KL}(\eta_t||\nu) = \mathrm{KL}(\mu_t||\pi)$ and $\left|\left|\nabla\left(\frac{\eta_t}{\nu}\right)\right|\right|_{L^2(\nu)}^2 = \left|\left|[\nabla^2\phi]^{-1}\nabla\left(\frac{\mu_t}{\pi}\right)\right|\right|_{L^2(\pi)}^2$, from which the conclusion follows. $\square$

**Proposition 12.** *The dissipation of $\chi^2(\cdot||\pi)$ along the MWGF in (66) is given by*

$$\frac{d\chi^2(\mu_t||\pi)}{dt} = -4\left|\left|[\nabla^2\phi]^{-1}\nabla\left(\frac{\mu_t}{\pi}\right)\right|\right|_{L^2(\mu_t)}^2. \tag{68}$$

*Proof.* Similar to the last result, the proof follows very closely the proof of Proposition 1. On this occasion, we have

$$\frac{d\chi^2(\eta_t||\nu)}{dt} = \left\langle w_t, 2\nabla\left(\frac{\eta_t}{\nu}\right)\right\rangle_{L^2(\eta_t)} = -4\left|\left|\nabla\left(\frac{\eta_t}{\nu}\right)\right|\right|_{L^2(\eta_t)}^2. \tag{69}$$

Arguing as in (39) - (40) and (41) - (43) in the proof of Proposition 2, we have that $\chi^2(\eta||\nu) = \chi^2(\mu||\pi)$ and $\left|\left|\nabla\left(\frac{\eta_t}{\nu}\right)\right|\right|_{L^2(\eta_t)}^2 = \left|\left|[\nabla^2\phi]^{-1}\nabla\left(\frac{\mu_t}{\pi}\right)\right|\right|_{L^2(\mu_t)}^2$, from which the conclusion follows. $\square$

Similar to before, we can also obtain exponential convergence to the target under a mirrored functional inequality.

**Proposition 13.** *Assume that $\pi$ satisfies a mirrored PI with constant $\kappa > 0$. In particular, for all locally Lipschitz $g \in L^2(\pi)$, there exists $\kappa > 0$ such that*

$$\mathrm{Var}_\pi[g] \leq \frac{1}{\kappa} \left\|[\nabla^2\phi]^{-1}\nabla g\right\|^2_{L^2(\pi)} \tag{70}$$

*Then the KL divergence converges exponentially along the MWGF in* (66)*:*

$$\mathrm{KL}(\mu_t||\pi) \leq \mathrm{KL}(\mu_0||\pi)e^{-2\lambda t}. \tag{71}$$

*Furthermore, for $t \geq \frac{1}{2\lambda}$, the following convergence rate holds:*

$$\mathrm{KL}(\mu_t||\pi) \leq \left(\mathrm{KL}(\mu_0||\pi) \wedge 2\right)e^{-2\lambda t}. \tag{72}$$

*Proof.* The proof follows closely the proof of [20, Theorem 1]. In particular, substituting the mirrored PI (70) with $g = \frac{\mu_t}{\pi} - 1$ into (67), we have that

$$\frac{\mathrm{dKL}(\mu_t||\pi)}{\mathrm{d}t} = -\left\|[\nabla^2\phi]^{-1}\nabla\left(\frac{\mu_t}{\pi}\right)\right\|^2_{L^2(\mu_t)} \leq -2\lambda\chi^2(\mu_t||\pi) \leq -2\lambda\mathrm{KL}(\mu_t||\pi), \tag{73}$$

where in the final inequality we have used the fact that $\mathrm{KL}(\cdot||\pi) \leq \chi^2(\cdot||\pi)$ (e.g., [93, Sec. 2.4]). The first bound now follows straightforwardly from Grönwall's inequality.

For the second bound, we will instead use the inequality $\mathrm{KL}(\cdot||\pi) \leq \log\left(1 + \chi^2(\cdot||\pi)\right)$ (e.g., [93, Sec. 2.4]), which implies that $e^{\mathrm{KL}(\cdot||\pi)} - 1 \leq \chi^2(\cdot||\pi)$. Using this in (73), it follows that

$$\frac{\mathrm{d}}{\mathrm{d}t}\left(1 - e^{-\mathrm{KL}(\mu_t||\pi)}\right) \leq -2\lambda\left(1 - e^{-\mathrm{KL}(\mu_t||\pi)}\right). \tag{74}$$

Applying Grönwall's lemma, we then have that

$$\left(1 - e^{-\mathrm{KL}(\mu_t||\pi)}\right) \leq e^{-2\lambda t}\left(1 - e^{-\mathrm{KL}(\mu_t||\pi)}\right) \leq e^{-2\lambda t}, \tag{75}$$

where in the final line we have used the elementary inequality $g(x) = 1 - e^{-x} \leq 1$, for $x \geq 0$. Finally, observe that if $t \geq \frac{1}{2\lambda}$, then $\left(1 - e^{-\mathrm{KL}(\mu_t||\pi)}\right) \leq e^{-2\lambda t} \leq e^{-1}$. Combining this with the fact that $x \leq 2g(x)$ whenever $g(x) \leq e^{-1}$, we have that

$$\mathrm{KL}(\mu_t||\pi) \leq 2\left(1 - e^{-\mathrm{KL}(\mu_t||\pi)}\right) \leq 2e^{-2\lambda t} \tag{76}$$

for $t \geq \frac{1}{2\lambda}$. Alongside the bound in (71), this completes the proof of (72). $\square$

### B.3.2 Kernelised MWGF of the Chi-Squared Divergence

In [20], the authors provide an interpretation of SVGD as a kernelised WGF of the $\chi^2$ divergence. Similarly, we can interpret MSVGD as a kernelised MWGF of the $\chi^2$ divergence; see also [82, App. H]. In particular, observe that [20, Sec. 2.3]

$$\mathcal{P}_{\eta_t,k_\phi}\nabla\log\left(\frac{\eta_t}{\nu}\right) = \int k(x,\cdot)\nabla\log\left(\frac{\eta_t}{\nu}(x)\right)\mathrm{d}\eta_t(x) \tag{77}$$

$$= \int k(x,\cdot)\nabla\left(\frac{\eta_t}{\nu}(x)\right)\mathrm{d}\nu(x) = \mathcal{P}_{\nu,k_\phi}\nabla\left(\frac{\eta_t}{\nu}\right). \tag{78}$$

It follows that the continuous-time MSVGD dynamics in (5) or (54) can equivalently be written as

$$\frac{\partial\eta_t}{\partial t} - \nabla\cdot\left(P_{\nu,k_\phi}\nabla\left(\frac{\eta_t}{\nu}\right)\eta_t\right) = 0, \quad \mu_t = (\nabla\phi^*)_\#\eta_t. \tag{79}$$

Comparing (79) and (66), it is clear that, up to a factor of two, MSVGD can be interpreted as the MWGF obtained by replacing the gradient of chi-squared divergence, $\nabla\left(\frac{\eta_t}{\nu}\right)$, by $P_{\nu,k_\phi}\nabla\left(\frac{\eta_t}{\nu}\right)$. In [82, App. H], the authors obtain continuous-time convergence results for this scheme, under a mirrored Stein PI. Here, we proceed differently, based on the approach in [20].

### B.3.3 Mirrored Laplacian Adjusted Wasserstein Gradient Descent (MLAWGD)

Following [20], suppose now that we replace $P_{\nu,k_\phi} \nabla \left( \frac{\eta_t}{\nu} \right)$ by the vector field $\nabla P_{\nu,k_\phi} \left( \frac{\eta_t}{\nu} \right)$. The new dynamics, which we refer to as the mirrored Laplacian adjusted Wasserstein gradient flow (MLAWGF), are then given by

$$\frac{\partial \eta_t}{\partial t} - \nabla \cdot \left( \nabla P_{\nu,k_\phi} \left( \frac{\eta_t}{\pi} \right) \eta_t \right) = 0, \quad \mu_t = (\nabla \phi^*)_\# \eta_t. \tag{80}$$

The dynamics in (80) can essentially be viewed as a change of measure of the standard LAWGD dynamics in [20, Sec. 4], designed to converge to the dual target $\nu = (\nabla \phi)_\# \pi$.

Suppose, in addition, that the kernel $k_\phi$ were chosen such that $P_{\nu,k_\phi} = -\mathcal{L}_\nu^{-1}$, where $\mathcal{L}_{\nu,k_\phi}$ denotes the infinitesimal generator of the Langevin SDE with stationary distribution $\nu$. Thus, in particular,

$$\mathcal{L}_\nu f_\phi(y) = - \langle \nabla W(y), \nabla f_\phi(y) \rangle + \Delta f_\phi(y). \tag{81}$$

where $f_\phi : \mathbb{R}^d \to \mathbb{R}$ denotes the function $f_\phi(\cdot) = f(\nabla \phi^*(\cdot))$, given some base function $f : \mathcal{X} \to \mathbb{R}$. Letting $y = \nabla \phi(x)$, using (20), and then the chain rule, we can rewrite this as

$$\mathcal{L}_v f(x) = - \langle [\nabla^2 \phi(x)]^{-1} [\nabla V(x) + \nabla \log \det \nabla^2 \phi(x)], [\nabla^2 \phi(x)]^{-1} \nabla f(x) \rangle \tag{82}$$

$$+ \operatorname{Tr} \left[ -[\nabla^2 \phi(x)]^{-1} \nabla^3 \phi(x) [\nabla^2 \phi(x)]^{-2} \nabla f(x) + [\nabla^2 \phi(x)]^{-2} \nabla^2 f(x) \right]. \tag{83}$$

We will denote this kernel $k_{\mathcal{L}_\nu}$. In practice, computing $k_{\mathcal{L}_\nu}$ requires computing the spectral decomposition of the operator $\mathcal{L}_\nu$, which is challenging even in the moderate dimensions. Nonetheless, following [20, Theorem 4], for this choice of kernel we can show that the MLAWGF converges exponentially fast to the target distribution, at a rate independent of the Poincaré constant.

**Proposition 14.** *Assume that $\pi$ satisfies a mirrored PI with some constant $\kappa > 0$. In addition, suppose that $\mathcal{L}_\nu$ has a discrete spectrum. Then the KL divergence converges exponentially along the MLAWGF, with a rate independent of $\kappa$:*

$$\mathrm{KL}(\mu_t || \pi) \leq (\mathrm{KL}(\mu_0 || \pi) \wedge 2) e^{-t}. \tag{84}$$

*Proof.* Using standard rules for calculus on the Wasserstein space, and the integration by parts formula $\mathbb{E}_\nu \langle \nabla f, \nabla g \rangle = \mathbb{E}_\nu [f \mathcal{L}_\nu g]$ [e.g., 5], we have

$$\frac{\mathrm{dKL}(\eta_t || \nu)}{\mathrm{d}t} = - \left\langle \nabla \left( \frac{\eta_t}{\nu} \right), \nabla P_{\nu,k_{\mathcal{L}_\nu}} \left( \frac{\eta_t}{\nu} \right) \right\rangle_{L^2(\pi)} = \int \frac{\eta_t}{\nu} \mathcal{L}_\nu P_{\nu,k_{\mathcal{L}_\nu}} \frac{\eta_t}{\nu} \mathrm{d}\nu(x) = - \left\| \frac{\eta_t}{\nu} \right\|_{L^2(\nu)}^2.$$

By [43, Theorem 2] and (41) - (43) in the proof of Proposition 2, we have that $\mathrm{KL}(\eta_t || \nu) = \mathrm{KL}(\mu_t || \pi)$ and $\left\| \nabla \left( \frac{\eta_t}{\nu} \right) \right\|_{L^2(\nu)}^2 = \left\| [\nabla^2 \phi]^{-1} \left( \frac{\mu_t}{\pi} \right) \right\|_{L^2(\pi)}^2$, which implies that

$$\frac{\mathrm{dKL}(\mu_t || \pi)}{\mathrm{d}t} = - \left\| \frac{\mu_t}{\pi} \right\|_{L^2(\pi)}^2 = - \left\| \left( \frac{\mu_t}{\pi} - 1 \right) \right\|_{L^2(\pi)}^2 = - \mathcal{X}^2(\mu_t || \pi) \leq - \mathrm{KL}(\mu_t || \pi), \tag{85}$$

where in the final line we have once again used the fact that $\mathrm{KL}(\cdot || \pi) \leq \mathcal{X}^2(\cdot || \pi)$. The conclusion now follows via Grönwall's inequality. $\qquad\square$

To obtain an implementable algorithm, observe that the vector field in the continuity equation can be rewritten as [20]

$$-\nabla P_{\nu,k_{\mathcal{L}_\nu}} \frac{\eta_t}{\nu}(\cdot) = - \int \nabla_1 k_{\mathcal{L}_\nu}(\cdot, y) \frac{\eta_t}{\nu}(y) \mathrm{d}\nu(y) = - \int \nabla_1 k_{\mathcal{L}_\nu}(\cdot, y) \mathrm{d}\eta(y). \tag{86}$$

Next, using a forward Euler discretisation in time, we arrive at

$$\eta_{t+1} = \left( \mathrm{id} - \gamma \nabla P_{\nu,k_{\mathcal{L}_\nu}} \frac{\eta_t}{\nu} \right)_\# \eta_t, \quad \mu_{t+1} = (\nabla \phi)_\# \eta_{t+1}. \tag{87}$$

Finally, we approximate the expectations in (87) with samples. In particular, after initialising a collection of particles $x_0^i \overset{\text{i.i.d.}}{\sim} \mu_0$ for $i \in [N]$, we set $y_0^i = \nabla \phi(x_0^i)$ for $i \in [N]$, and then update

$$y_{t+1}^i = y_t^i - \gamma \frac{1}{N} \sum_{j=1}^N \nabla_{y^i} k_{\mathcal{L}_\nu}(y_t^i, y_t^j), \quad x_{t+1}^i = \nabla \phi^*(y_{t+1}^i). \tag{88}$$

This algorithm, which we refer to as the mirrored LAWGD (MLAWGD) algorithm, is summarised in Alg. 4.

**Algorithm 4** Mirrored LAWGD

---

**input:** target density $\pi$, kernel $k_{\mathcal{L}_\nu}$, mirror function $\phi$, particles $(x_0^i)_{i=1}^N \sim \mu$, step size $\gamma$.
**initialise:** for $i \in [N]$, $y_0^i = \nabla\phi(x_0^i)$.
**for** $t = 0, \ldots, T-1$ **do**
    For $i \in [N]$, $y_{t+1}^i = y_t^i - \gamma \frac{1}{N} \sum_{j=1}^N \nabla_{y^i} k_{\mathcal{L}_\nu}(y_t^i, y_t^j)$.
    For $i \in [N]$, $x_{t+1}^i = \nabla\phi^*(y_{t+1}^i)$.
**return** $(x_T^i)_{i=1}^N$.

---

## B.4 Mirrored Kernel Stein Discrepancy Descent

Finally, we consider the MWGF of the kernel Stein discrepancy (KSD) [21, 38, 62]. In so doing, we will obtain a mirrored version of the kernel Stein discrepancy descent (KSDD) algorithm in [51].

### B.4.1 MWGF of the Kernel Stein Discrepancy

We begin by defining $\mathcal{F}(\eta) = \frac{1}{2}\mathrm{KSD}_\phi^2(\eta|\nu)$, where $\mathrm{KSD}_\phi(\eta|\nu)$ is the *mirrored KSD*, which we define as

$$\mathrm{KSD}_\phi(\eta|\nu) = \sqrt{\int_{\mathbb{R}^d}\int_{\mathbb{R}^d} k_{\nu,\phi}(y,y')\mathrm{d}\eta(y)\mathrm{d}\eta(y')}, \tag{89}$$

where $k_{\nu,\eta}$ is the mirrored Stein kernel, defined in terms of the score $s_\nu = \nabla\log\nu$ of the mirrored target $\nu = (\nabla\phi)_{\#}\pi$, and the positive semi-definite kernel $k_\phi$ as

$$\begin{aligned} k_{\nu,\phi}(y,y') &= \nabla_y\log\nu(y)^\top k_\phi(y,y')\nabla_{y'}\log\nu(y') + \nabla_y\log\nu(y)^\top\nabla_{y'}k_\phi(y,y') \\ &\quad + \nabla_y k_\phi^\top(y,y')\nabla\log\nu(y') + \langle\nabla_y k_\phi(y,\cdot), \nabla_{y'}k_\phi(\cdot,y')\rangle_{\mathcal{H}_{k_\phi}^d}, \end{aligned} \tag{90}$$

and where, as before, $k_\phi$ is the kernel defined according to $k_\phi(y,y') = k(\nabla\phi^*(y), \nabla\phi^*(y'))$, for some base kernel $k : \mathcal{X} \times \mathcal{X} \to \mathbb{R}$. The MKSD is nothing more than the KSD between $\eta = (\nabla\phi)_{\#}\mu$ and $\nu = (\nabla\phi)_{\#}\pi$, with respect to the kernel $k_\phi$.

In this case, one can show that $\nabla_{W_2}\mathcal{F}(\eta) = \mathbb{E}_{y\sim\eta}[\nabla_2 k_{\nu,\phi}(y,\cdot)]$ [51, Proposition 2]. We can thus define the MKSD gradient flow according to

$$\frac{\partial\eta_t}{\partial t} - \nabla\cdot(\mathbb{E}_{y\sim\eta_t}[\nabla_2 k_{\nu,\phi}(y,\cdot)]\eta_t) = 0, \quad \mu_t = (\nabla\phi^*)_{\#}\eta_t. \tag{91}$$

Similar to the previous sections, we would like to study the properties of this scheme in continuous time.

**Proposition 15.** *The dissipation of $\frac{1}{2}\mathrm{KSD}_\phi^2(\eta_t|\nu)$ along the MWGF in* (91) *is given by*

$$\frac{\mathrm{d}\frac{1}{2}\mathrm{KSD}_\phi^2(\eta_t|\nu)}{\mathrm{d}t} = -\mathbb{E}_{y'\sim\eta_t}\left[||\mathbb{E}_{y\sim\eta_t}[\nabla_{y'}k_{\nu,\phi}(y,y')]||^2\right] \tag{92}$$

*Proof.* The result follows straightforwardly using the chain rule. In particular, we have

$$\frac{\mathrm{d}\frac{1}{2}\mathrm{KSD}_\phi^2(\eta_t|\nu)}{\mathrm{d}t} = \left\langle w_t, \nabla_{W_2}\frac{1}{2}\mathrm{KSD}^2(\eta_t|\nu)\right\rangle_{L^2(\eta_t)} \tag{93}$$

$$= -\mathbb{E}_{y'\sim\eta_t}\left[||\mathbb{E}_{y\sim\eta_t}[\nabla_{y'}k_{\nu,\phi}(y,y')]||^2\right]. \tag{94}$$

$\square$

**Remark 10.** *Unlike elsewhere (i.e., Propositions 1 - 2, Proposition 8, and Propositions 11 - 12) this dissipation result is given in terms of the mirrored densities $(\eta_t)_{t\geq 0}$ and the mirrored target $\nu = (\nabla\phi)_{\#}\pi$, rather than densities $(\mu_t)_{t\geq 0}$ and the target $\pi$. The crucial difference here is that the objective functional $\frac{1}{2}\mathrm{KSD}^2(\eta_t|\nu)$ is not equal to $\frac{1}{2}\mathrm{KSD}^2(\mu_t|\pi)$, whereas previously it was true that, e.g., $\mathrm{KL}(\eta_t||\nu) = \mathrm{KL}(\mu_t||\pi)$ or $\chi^2(\eta_t||\nu) = \chi^2(\mu_t||\pi)$. Nonetheless, it is possible to give a dissipation result in terms of the target $\pi$.*

**Algorithm 5** Mirrored KSDD

---

**input:** target density $\pi$, kernel $k$, mirror function $\phi$, particles $(x_0^i)_{i=1}^N \sim \mu$, step size $\gamma$.
**initialise:** for $i \in [N]$, $y_0^i = \nabla\phi(x_0^i)$.
**for** $t = 0, \ldots, T-1$ **do**
    For $i \in [N]$, $y_{t+1}^i = y_t^i - \gamma\frac{1}{N^2}\sum_{j=1}^N \nabla_{y^i}k_{\nu,\phi}(y_t^j, y_t^i)$.
    For $i \in [N]$, $x_{t+1}^i = \nabla\phi^*(y_{t+1}^i)$.
**return** $(x_T^i)_{i=1}^N$.

---

*We begin by rewriting the LHS of* (92). *Recall that the squared KSD between $\mu$ and $\pi$ is identical to the Stein Fisher information between $\mu$ and $\pi$ [e.g.,* 62, *Theorem 3.6]. This also holds for $\eta = (\nabla\phi)_{\#}\mu$ and $\nu = (\nabla\phi)_{\#}\pi$ in the mirrored space, and thus we have*

$$\text{KSD}_\phi^2(\eta|\nu) = \left\|\mathcal{S}_{\mu,k_\phi}\nabla\log\left(\frac{\eta}{\nu}\right)\right\|_{\mathcal{H}_{k_\phi}^d}^2 = \left\|S_{\mu,k}[\nabla^2\phi]^{-1}\nabla\log\left(\frac{\mu}{\pi}\right)\right\|_{\mathcal{H}_k^d}^2, \tag{95}$$

*where in the second equality we have used the result obtained in* (57) - (60); *see the proof of Proposition* 8. *Thus, in particular, $\mathcal{F}(\eta)$ is equal to the mirrored Stein Fisher information, up to a factor half.*

*We now turn our attention to the RHS of* (92). *We would like to simplify $\nabla_{y'}k_{\nu,\phi}(y, y')$, as defined in* (90). *Let $y = \nabla\phi(x)$ and $y' = \nabla\phi(x')$. Using the definition of $k_\phi$, we have $k_\phi(y, y') = k(x, x')$. Using also the chain rule, we have*

$$\nabla_y k_\phi(y, y') = [\nabla^2\phi(x)]^{-1}\nabla_x k(x, x') \quad, \quad \nabla_{y'}k_\phi(y, y') = [\nabla^2\phi(x')]^{-1}\nabla_{x'}k(x, x') \tag{96}$$

*Similarly, it holds that*

$$\langle\nabla_y k_\phi(y, \cdot), \nabla_{y'}k_\phi(\cdot, y')\rangle_{\mathcal{H}_{k_\phi}^d} = \langle[\nabla^2\phi(x)]^{-1}\nabla_x k(x, \cdot), [\nabla^2\phi(x')]^{-1}\nabla_{x'}k(\cdot, x')\rangle_{\mathcal{H}_k^d}. \tag{97}$$

*Finally, due to* (20), *we have $\nabla_y\log\nu(y) = [\nabla^2\phi(x)]^{-1}[\nabla_x\log\pi(x) - \nabla_x\log\det\nabla^2\phi(x)]$. Putting everything together, we thus have that*

$$
\begin{aligned}
k_{\nu,\phi}(y, y') = &\left([\nabla^2\phi(x)]^{-1}[\nabla_x\log\pi(x) - \nabla_x\log\det\nabla^2\phi(x)]\right)^\top k(x, x') \\
&\cdot \left([\nabla^2\phi(x)]^{-1}[\nabla_x\log\pi(x) - \nabla_x\log\det\nabla^2\phi(x)]\right) \\
&+ \left([\nabla^2\phi(x)]^{-1}[\nabla_x\log\pi(x) - \nabla_x\log\det\nabla^2\phi(x)]\right)^\top\left([\nabla^2\phi(x)]^{-1}\nabla_x k(x, x')\right) \\
&+ \left([\nabla^2\phi(x')]^{-1}\nabla_{x'}k(x, x')\right)^\top\left([\nabla^2\phi(x)]^{-1}[\nabla_x\log\pi(x) - \nabla_x\log\det\nabla^2\phi(x)]\right) \\
&+ \langle[\nabla^2\phi(x)]^{-1}\nabla_x k(x, \cdot), [\nabla^2\phi(x')]^{-1}\nabla_{x'}k(\cdot, x')\rangle := \tilde{k}_{\pi,\phi}(x, x').
\end{aligned} \tag{98}
$$

*Using* (95) *and* (98) *we thus have the following dissipation result for the mirrored Stein Fisher information in terms of the target $\pi$:*

$$\frac{d}{dt}\left\|S_{\mu_t,k}[\nabla^2\phi]^{-1}\nabla\log\left(\frac{\mu_t}{\pi}\right)\right\|_{\mathcal{H}_k^d}^2 = -2\mathbb{E}_{x'\sim\mu_t}\left[\left\|\mathbb{E}_{x\sim\mu_t}\left[\tilde{k}_{\pi,\phi}(x, x')\right]\right\|^2\right]. \tag{99}$$

### B.4.2 Mirrored Kernel Stein Discrepancy Descent (MKSDD)

To obtain an implementable MKSDD algorithm, it is, of course, necessary to discretise in time and in space. In this case, following [51], we will first discretise in space, replacing the measure $\eta$ by a discrete approximation $\hat{\eta} = \frac{1}{N}\sum_{j=1}^N \delta_{y^j}$. This yields, writing $\omega^N = \{y^1, \ldots, y^n\}$,

$$F(\omega^N) := \mathcal{F}(\hat{\eta}) = \frac{1}{2N^2}\sum_{i,j=1}^N k_{\nu,\phi}(y^i, y^j), \quad \nabla_{y^i}F(\omega^N) := \mathcal{F}(\hat{\eta}) = \frac{1}{N^2}\sum_{j=1}^N \nabla_2 k_{\nu,\phi}(y^j, y^i). \tag{100}$$

The mirrored KSDD algorithm thus takes the following form. Initialise a collection of particles $x_0^i \overset{\text{i.i.d.}}{\sim} \mu_0$ for $i \in [N]$, and set $y_0^i = \nabla\phi(x_0^i)$ for $i \in [N]$. Then, update

$$y_{t+1}^i = y_t - \gamma\frac{1}{N^2}\sum_{j=1}^N \nabla_2 k_{\nu,\phi}(y_t^j, y_t^i) \quad, \quad x_{t+1}^i = \nabla\phi^*(y_{t+1}^i). \tag{101}$$

## C Convergence of Mirrored Coin Wasserstein Gradient Descent

**Proposition 16.** *Suppose that $\pi$ is strongly log-concave and that $||\nabla_{W_2}\mathcal{F}(\eta_t)(y_t)|| \leq L$ for all $t \in [T]$. In addition, suppose that $(\eta_t)_{t\in\mathbb{N}_0}$ and $\nu$ satisfy [81, Assumption B.1]. Then there exists a mirror map $\phi$ such that*

$$\mathrm{KL}\left(\frac{1}{T}\sum_{t=1}^{T}\mu_t \middle\| \pi\right) \leq \frac{L}{T}\left[1 + \int_{\mathcal{X}} ||\nabla\phi(x)|| \sqrt{T\ln\left(1 + 96K^2T^2||\nabla\phi(x)||^2\right)}\pi(\mathrm{d}x)\right.$$
$$\left. + \int_{\mathcal{X}} ||\nabla\phi(x)|| \sqrt{T\ln\left(1 + 96T^2||\nabla\phi(x)||^2\right)}\mu_0(\mathrm{d}x)\right], \quad (102)$$

*where $K > 0$ is the constant defined in [81, Assumption B.1].*

*Proof.* First note, using Jensen's inequality, and [43, Theorem 2], that

$$\mathrm{KL}\left(\frac{1}{T}\sum_{t=1}^{T}\mu_t \middle\| \pi\right) \leq \frac{1}{T}\sum_{t=1}^{T}\mathrm{KL}\left(\mu_t||\pi\right) \tag{103}$$

$$= \frac{1}{T}\sum_{t=1}^{T}\left[\mathrm{KL}(\eta_t||\nu) - \mathrm{KL}(\nu||\nu)\right] \tag{104}$$

$$:= \frac{1}{T}\sum_{t=1}^{T}\mathcal{F}(\eta_t) - \mathcal{F}(\nu), \tag{105}$$

where in the final line we have defined $\mathcal{F}(\eta) := \mathrm{KL}(\eta||\pi)$. Now, under the assumption that $\pi$ is strongly log-concave, [43, Proof of Theorem 3] guarantees the existence of a mirror map $\phi : \mathcal{X} \to \mathbb{R} \cup \{\infty\}$ such that the mirrored target $\nu = (\nabla\phi)_{\#}\pi$ is also strongly log-concave. It follows, in particular, that $\mathcal{F}(\eta) := \mathrm{KL}(\eta||\pi)$ is geodesically convex [e.g., 2, Chapter 9]. Thus, arguing as in the proof of [81, Proposition 3.3], with $w_0 = 1$, it follows that

$$\frac{1}{T}\sum_{t=1}^{T}\mathcal{F}(\eta_t) - \mathcal{F}(\nu) \leq \frac{L}{T}\left[1 + \int_{\mathbb{R}^d} ||y|| \sqrt{T\ln\left(1 + 96K^2T^2||y||^2\right)}\nu(\mathrm{d}y)\right.$$
$$\left. + \int_{\mathbb{R}^d} ||y|| \sqrt{T\ln\left(1 + 96T^2||y||^2\right)}\eta_0(\mathrm{d}y)\right]. \tag{106}$$

Finally, once more using the definition of the mirror map and in particular the fact that $\nu = (\nabla\phi)_{\#}\pi$ and $\eta_0 = (\nabla\phi)_{\#}\eta_0$, we can rewrite the RHS in (106) as the RHS in (102). This completes the proof. $\square$

## D Other Mirrored Coin Sampling Algorithms

In this section, we provide some other mirrored coin sampling algorithms besides Coin MSVGD. Recall, from Sec. 4.2, the general form of the mirrored coin sampling algorithm. Let $x_0 \sim \mu_0 \in \mathcal{P}_2(\mathcal{X})$, and $y_0 = \nabla\phi(x_0) \in \mathbb{R}^d$. Then, for $t \in \mathbb{N}$, update

$$y_t = y_0 + \frac{\sum_{s=1}^{t-1}c_{\eta_s}(y_s)}{t}\left(1 + \sum_{s=1}^{t-1}\langle c_{\eta_s}(y_s), y_s - y_0\rangle\right) \quad, \quad x_t = \nabla\phi^*(x_t), \tag{107}$$

where $c_{\eta_t}(y_t) = -\nabla_{W_2}\mathcal{F}(\eta_t)(y_t)$, or some variant thereof. As noted in Sec. 4.2, for different choices of $\mathcal{F}$ and $(c_{\eta_t})_{t\in\mathbb{N}_0}$, one obtains learning-rate free analogues of different mirrored ParVI algorithms.

In Sec. 4.2, we provided the Coin MSVGD algorithm (Alg. 2), which corresponds to $\mathcal{F} = \mathrm{KL}(\cdot|\nu)$ and $(c_{\eta_t})_{t\in\mathbb{N}_0} = (-P_{\eta_t,k_\phi}\nabla\log(\frac{\eta_t}{\nu}))_{t\in\mathbb{N}_0}$. This can be viewed as the coin sampling analogue of MSVGD (Sec. 3.2.1 and App. B.2). We now provide two more mirrored coin sampling algorithms, which correspond to the coin sampling analogues of MLAWGD (App. B.3) and MKSDD (App. B.4).

## D.1 Coin MLAWGD

---

**Algorithm 6** Coin MLAWGD

---

**input:** target density $\pi$, kernel $k_{\mathcal{L}_\nu}$, mirror map $\phi$, initial particles $(x_0^i)_{i=1}^N \sim \mu_0$.
**initialise:** for $i \in [N]$, $y_0^i = \nabla\phi(x_0^i)$.
**for** $t = 1, \ldots, T$ **do**
    For $i \in [N]$,

$$y_t^i = y_0^i - \frac{\sum_{s=1}^{t-1} \frac{1}{N} \sum_{j=1}^N \nabla_{y^i} k_{\mathcal{L}_\nu}(y_s^i, y_s^j)}{t} \left(1 - \sum_{s=1}^{t-1} \langle \frac{1}{N} \sum_{j=1}^N \nabla_{y^i} k_{\mathcal{L}_\nu}(y_s^i, y_s^j), y_s^i - y_0^i \rangle \right)$$

$$x_t^i = \nabla\phi^*(y_t^i).$$

where $k_{\mathcal{L}_\nu}$ is the mirrored LAWGD kernel defined in (81).
**return** $(x_T^i)_{i=1}^N$.

---

## D.2 Coin MKSDD

---

**Algorithm 7** Coin MKSDD

---

**input:** target density $\pi$, kernel $k$, mirror map $\phi$, initial particles $(x_0^i)_{i=1}^N \sim \mu_0$.
**initialise:** for $i \in [N]$, $y_0^i = \nabla\phi(x_0^i)$.
**for** $t = 1, \ldots, T$ **do**
    For $i \in [N]$

$$y_t^i = y_0^i - \frac{\sum_{s=1}^{t-1} \frac{1}{N^2} \sum_{j=1}^N \nabla_{y^i} k_{\nu,\phi}(y_s^j, y_s^i)}{t} \left(1 - \sum_{s=1}^{t-1} \langle \frac{1}{N^2} \sum_{j=1}^N \nabla_{y^i} k_{\nu,\phi}(y_s^j, y_s^i), y_s^i - y_0^i \rangle \right)$$

$$x_t^i = \nabla\phi^*(y_t^i).$$

where $k_{\nu,\phi}$ is the mirrored Stein kernel defined in (90).
**return** $(x_T^i)_{i=1}^N$.

---

# E    Adaptive Coin MSVGD

In Sec. 4.1 - 4.2, we described a betting strategy which is valid under the assumption that the sequence of outcomes $(c_t)_{t\in\mathbb{N}} := (c_{\eta_t}(y_t))_{t\in\mathbb{N}}$ are bounded by one, in the sense that $||c_{\eta_t}(y_t^i)|| \leq 1$ for all $t \in [T]$ and for all $i \in [N]$ [see also 81, Sec. 3.4]. In practice, this may not be the case. If, instead, $||c_{\eta_t}(y_t^i)|| \leq L$ for some constant $L$, then one can simply replace $c_{\eta_t}(y_t)$ by its normalised version, namely $\hat{c}_{\eta_t}(y_t) = c_{\eta_t}(y_t)/L$ in Alg. 2.

If, on the other hand, the constant $L$ is unknown in advance, then one can use a coordinate-wise empirical estimate $L_{t,j}^i$ which is adaptively updated as the algorithm runs, following [72] and later [81, App. D].[3] We provide full details of this version of Coin MSVGD in Alg. 8. One can also obtain an analogous version of Coin MIED; we omit the full details here in the interest of brevity.

Finally, when we use a coin sampling algorithm to perform inference for a (fairness) Bayesian neural network (see Sec. 6.3), we alter the denominator of the betting fraction in Alg. 8 such that it is at least $100L_{t,j}^i$. Thus, the update equation in Alg. 8 (or, analogously, the update equation in the adaptive version of Coin MIED) now reads as

$$y_{t,j}^i = y_{0,j}^i + \frac{\sum_{s=1}^{t-1} c_{s,j}^i}{\max(G_{t,j}^i + L_{t,j}^i, 100L_{t,j}^i)}(1 + \frac{R_{t,j}^i}{L_{t,j}^i}). \tag{108}$$

This modification, recommended in [72, Sec. 6], has the effect of restricting the value of the particles in the first few iterations. We should emphasise that these adaptive algorithms, which in practice we use in all experiments, are still free of any hyperparameters which need to be tuned by hand.

---

[3]Here, $i \in [N]$ indexes the particle, and $j \in [d]$ indexes the dimension (see Alg. 8).

---
**Algorithm 8** Adaptive Coin MSVGD
---

**input:** target density $\pi$, kernel $k$, mirror map $\phi$, initial particles $(x_0^i)_{i=1}^N \sim \mu_0$.
**initialise:** for $i \in [N]$, $y_0^i = \nabla\phi(x_0^i)$; for $i \in [N]$ and $j \in [d]$, $L_{0,j}^i = 0$, $G_{0,j}^i = 0$, $R_{0,j}^i = 0$.
**for** $t = 1$ **to** $T$ **do**
    For $i \in [N]$,

$$c_{t-1}^i = -P_{\hat{\eta}_t, k_\phi} \nabla \log\left(\frac{\hat{\eta}_{t-1}}{\nu}\right)(y_{t-1}^i) \,,\ \hat{\eta}_{t-1} = \frac{1}{N}\sum_{k=1}^N \delta_{y_{t-1}^k} \qquad \text{(compute SVGD gradient)}$$

    For $i \in [N]$, $j \in [d]$,

$$L_{t,j}^i = \max(L_{t-1,j}^i, |c_{t-1,j}^i|) \qquad\qquad\qquad \text{(update max. observed scale)}$$
$$G_{t,j}^i = G_{t-1,j}^i + |c_{t-1,j}^i| \qquad\qquad\qquad \text{(update sum of abs. value of gradients)}$$
$$R_{t,j}^i = \max(R_{t-1,j}^i + \langle c_{t-1,j}^i, y_{t-1,j}^i - y_{0,j}^i\rangle, 0) \qquad \text{(update total reward)}$$
$$y_{t,j}^i = y_{0,j}^i + \frac{\sum_{s=1}^{t-1} c_{s,j}^i}{G_{t,j}^i + L_{t,j}^i}(1 + \frac{R_{t,j}^i}{L_{t,j}^i}). \qquad\qquad \text{(update the particles)}$$

    For $i \in [N]$, $x_t^i = \nabla\phi^*(y_t^i)$.
**output:** $(x_T^i)_{i=1}^N$.

---

**Computational Cost**. It is worth noting that, in terms of computational and memory cost, the adaptive Coin MSVGD algorithm is no worse than MSVGD when the latter is paired, as is common, with a method such as Adagrad [31], RMSProp [92], or Adam [48]. The computational cost per iteration is $O(N^2)$, due to the kernelised gradient. Meanwhile, in terms of memory requirements, it is necessary to keep track of the maximum observed absolute value of the gradients (component-wise), the sum of the absolute value of the gradients (component-wise), and the total reward (component-wise). This is similar to, e.g., Adam, which keeps track of exponential moving averages of the gradient and the (component-wise) squared gradient [48].

## F   Additional Experimental Details

In this section, we provide additional details relevant to the experiments carried out in Sec. 6. We implement all methods using Python 3, PyTorch, and TensorFlow. We use the existing implementations of MSVGD and SVMD [82] and MIED [60] to run these methods. We perform all experiments using a MacBook Pro 16" (2021) laptop with Apple M1 Pro chip and 16GB of RAM.

For the experiments in Sec. 6.1 and 6.2, we use the inverse multiquadric (IMQ) kernel $k(x, x') = (1 + \{||x - x'||_2^2/h^2)^{-\frac{1}{2}}$ due to its favourable convergence control properties (e.g., [38]). Unless otherwise stated, the bandwidth $h$ is determined using the median heuristic [36, 63]. For the experiment in Sec. 6.3, following [60], we use the $s$-Riesz family of mollififiers $\{\phi_\epsilon^s\}$ with $s = d + 10^{-4}$ and $\epsilon = 10^{-8}$.

### F.1   Simplex Targets

For both experiments in Sec. 6.1, we initialise all methods with i.i.d samples from a $\text{Dirichlet}(5)$ distribution and use the entropic mirror map

$$\phi(x) = \sum_{k=1}^d x_k \log x_k + (1 - \sum_{k=1}^d x_k)\log(1 - \sum_{k=1}^d x_k). \qquad (109)$$

We run all methods for $T = 500$ iterations using $N = 50$ particles. In the interest of a fair comparison, we run all of the learning-rate dependent methods (MSVGD, SVMD, projected SVGD) using RMSProp [92], an extension of the Adagrad algorithm [31], to adapt the learning rates.

**Sparse Dirichlet Posterior**. We first consider the sparse Dirichlet posterior in [73], which is given by

$$\pi(x) = \frac{1}{Z} \prod_{k=1}^{d+1} x_k^{n_k + \alpha_k - 1}, \quad x \in \text{int}(\Delta), \tag{110}$$

for parameters $\alpha_1, \ldots, \alpha_{d+1} > 0$ and observations $n_1, \ldots, n_{d+1}$, where $n_k$ is the number of observations of category $k$. We consider the sparse regime, in which most of the $n_k$ are zero and the remaining $n_k$ are large, e.g., of order $O(d)$. In particular, as in [82], we set $\alpha_k = 0.1$ for $k \in \{1, \ldots, 20\}$ and use count data $n_1 = 90$, $n_2 = n_3 = 5$, and $n_j = 0$ for $j > 3$. This target is used to replicate the multimodal sparse conditionals that arise in Latent Dirichlet Allocation (LDA) [9]. While this distribution is not log-concave, the dual-target is strictly log-concave under the entropic mirror map.

**Quadratic Target**. We next consider the quadratic simplex target introduced in [1]. In this case, the target takes the form

$$\pi(x) = \frac{1}{Z} \exp\left(-\frac{1}{2\sigma^2} x^\top A x\right), \tag{111}$$

where $A \in \mathbb{R}^{d \times d}$ is a symmetric positive semidefinite matrix with all entries bounded in magnitude by 1. Following [82], we set $\sigma = 0.01$ and generate $A$ by normalising products of random matrices with i.i.d. elements drawn from $\text{Unif}[-1, 1]$.

### F.2 Confidence Intervals for Post Selection Inference

Given data $(X, y) \in \mathbb{R}^{n \times p} \times \mathbb{R}^n$, the randomised Lasso is defined as the solution of [89, Sec. 3.1]

$$\hat{\beta} = \arg\min_{\beta \in \mathbb{R}^p} \left[ \frac{1}{2} ||y - X\beta||_2^2 + \lambda ||\beta||_1 - \omega^\top \beta + \frac{\varepsilon}{2} ||\beta||_2^2 \right], \tag{112}$$

where $\lambda$ is the $\ell_1$ penalty parameter, $\varepsilon \in \mathbb{R}_+$ is a ridge term, and $\omega \in \mathbb{R}^p$ is an auxiliary random vector drawn from some distribution $\mathbb{G}$ with density $g$. We choose $\mathbb{G}$ to be a zero-mean independent Gaussian distribution. The scale of this distribution, and the parameter $\varepsilon$, are both determined automatically by the `randomizedLasso` function in the `selectiveInference` package. The distribution of interest has density proportional to (e.g., [90, Sec. 4.2])

$$\hat{g}(\hat{\beta}_E, \hat{z}_{-E}) \propto g\left( \varepsilon (\hat{\beta}_E, 0)^\top - X^\top (y - X_E \hat{\beta}_E) + \lambda (\hat{z}_E, \hat{z}_{-E})^\top \right), \tag{113}$$

with support $\mathcal{D} = \{(\hat{\beta}_E, \hat{z}_{-E}) : \text{sign}(\hat{\beta}_E) = \hat{z}_E , ||\hat{z}_{-E}||_\infty \leq 1\}$. In all experiments, following [89], we integrate out $\hat{z}_{-E}$ analytically and reparameterise $\hat{\beta}_E$ as $\hat{z}_E \odot |\hat{\beta}_E|$. This yields a log-concave density for $|\hat{\beta}_E|$, with domain given by the non-negative orthant in $\mathbb{R}^q$. We then use the mirror map

$$\phi(x) = \sum_{k=1}^{d} (x_k \log x_k - x_k). \tag{114}$$

**Synthetic Example**. We consider the simulation in [79, Sec. 5.3], with $n = 100$ and $p = 40$. The design matrix $X$ is generated from an equi-correlated model. In particular, for $i = 1, \ldots, n$, the rows $x_i \overset{\text{i.i.d.}}{\sim} \mathcal{N}(0, \Sigma)$, where $\Sigma_{ij} = \delta_{i,j} + \rho(1 - \delta_{i,j})$, with $\rho = 0.3$. We also normalise the columns of $X$ to have norm 1. Meanwhile, the response is distributed as $y \sim \mathcal{N}(0, I_n)$, independent of $X$. That is the true parameter $\beta = 0$. We set $\varepsilon = \frac{\hat{\sigma}^2}{\sqrt{n}}$, where $\hat{\sigma}$ denotes the empirical standard deviation of the response. We set $\lambda$ using the default value returned by `theoretical.lambda` in the `selectiveInference` package, multiplied by a factor of $0.7n$. This adjustment reduces the $\ell_1$ regularisation, ensuring that a reasonable subset of the $p = 40$ features are selected.

We run each algorithm for $T = 1000$ iterations. For Coin MSVGD, MSVGD, and SVMD, we use $N = 50$ particles, and generate $N_{\text{total}}$ samples by aggregating the particles from $N_{\text{total}}/N$ independent runs. For the `norejection` method, we generate $N_{\text{total}}$ samples by running the algorithm for $N_{\text{total}}$ iterations after burn-in. Finally, following [82], in Fig. 3 the error bars for the actual coverage levels correspond to 95% Wilson intervals [101].

**HIV-1 Drug Resistance**. We use the vitro measurement of log-fold change under 3TC as the response and include mutations that appear more than 10 times in the dataset as predictors. The resulting dataset consists of $n = 633$ cases and $p = 91$ predictors. In the selection stage, we apply the randomised Lasso, setting the regularisation parameter to the theoretical value in `selectiveInference`, multiplied by a factor of $n$. We implement SVMD and MSVGD using RMSProp [92], with an initial learning rate of $\gamma = 0.01$ following [82]. We implement each method for $T = 2000$ iterations, using $N = 50$ particles.

### F.3 Fairness Bayesian Neural Network

We use a train-test split of 80% / 20%, as in [60, 65]. We implement the methods in [60, 65] using their source code, using the default hyperparameters. We run all algorithms for $T = 2000$ iterations, and using $N = 50$ particles. Similar to [60], we found that one of the algorithms (Control + SVGD) in [64] got stuck after initialisation, with accuracy close to $0.75$ and did not improve during training. We thus omit the results for this method in Fig. 5.

## G Additional Numerical Results

### G.1 Simplex Targets

In Fig. 6 and Fig. 7, we provide additional results for the sparse Dirichlet posterior [73] and the quadratic simplex target [1] considered in Sec. 6.1. In particular, in Fig. 6a and Fig. 7a, we plot the energy distance to a set of ground truth samples either obtained i.i.d. (sparse Dirichlet target) or using NUTS [42] (quadratic target) versus the number of iterations for Coin MSVGD, MSVGD, SVMD, and projected versions of Coin SVGD and SVGD. Meanwhile, in Fig. 6b and Fig. 7b, we compare the performance of Coin MSVGD with the performance of MSVGD, for multiple choices of the learning rate.

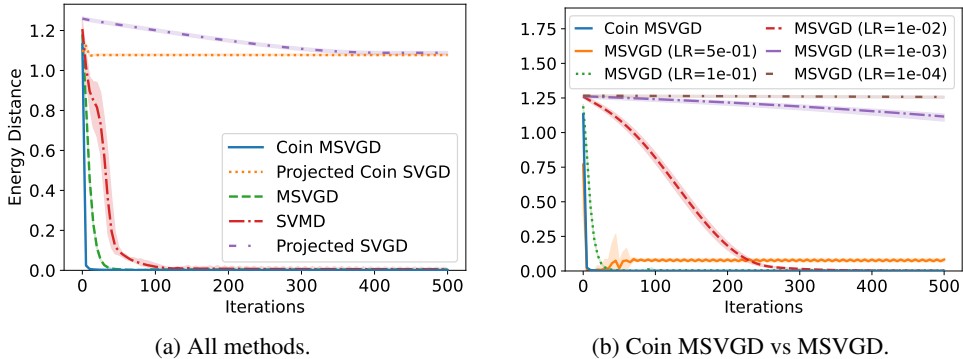

(a) All methods.                    (b) Coin MSVGD vs MSVGD.

Figure 6: **Additional results for the sparse dirichlet posterior in [73]**. (a) Energy distance vs. iterations for Coin MSVGD, MSVGD, SVMD, projected Coin SVGD, and projected SVGD; and (b) energy distance vs iterations for Coin MSVGD and MSVGD, using five different learning rates.

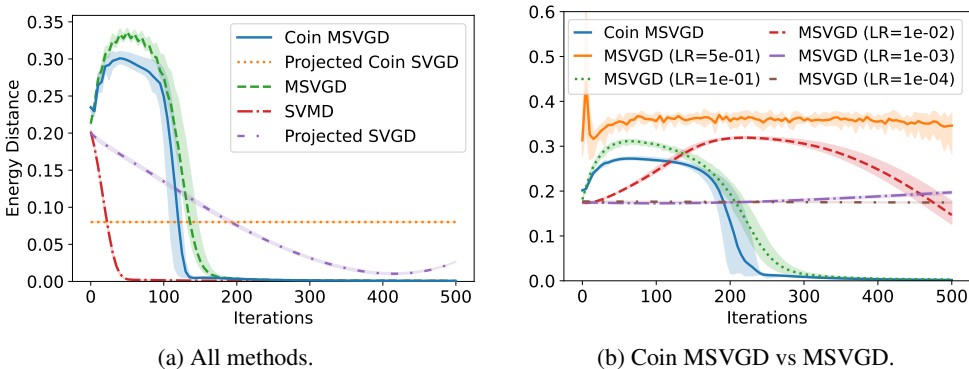

(a) All methods.                    (b) Coin MSVGD vs MSVGD.

Figure 7: **Additional results for the quadratic simplex target in [1]**. (a) Energy distance vs. iterations for Coin MSVGD, MSVGD, SVMD, projected Coin SVGD, and projected SVGD; and (b) energy distance vs iterations for Coin MSVGD and MSVGD, using five different learning rates.

As observed in Sec. 6.1, for the sparse Dirichlet posterior, Coin MSVGD converges more rapidly than MSVGD and SVMD, even for well-chosen values of the learning rate (Fig. 6). Meanwhile, for the quadratic target, Coin MSVGD generally converges more rapidly than MSVGD but not as fast as SVMD [82], which leverages the log-concavity of the target in the primal space (Fig. 7). In both cases, for poorly chosen values of the learning rate, MSVGD (e.g., $\gamma = 5 \times 10^{-1}$ in Fig. 6b and $\gamma = 1 \times 10^{-4}$, $\gamma = 1 \times 10^{-3}$, or $\gamma = 5 \times 10^{-1}$ in Fig. 7b) entirely fails to converge to the true target.

**Results for Different Numbers of Particles**. In Fig. 8 and Fig. 9, we provide additional results for the two simplex targets, now varying the number of particles $N$. The results are similar to those reported in Sec. 6.1. In particular, Coin MSVGD has comparable performance to MSVGD and SVMD with optimal learning rates, but significantly outperforms both algorithms for sub-optimal learning rates. In addition, as expected, the posterior approximation quality of Coin MSVGD, as measured by the energy distance, improves as a function of the number of particles $N$ (Fig. 8d and Fig. 9d).

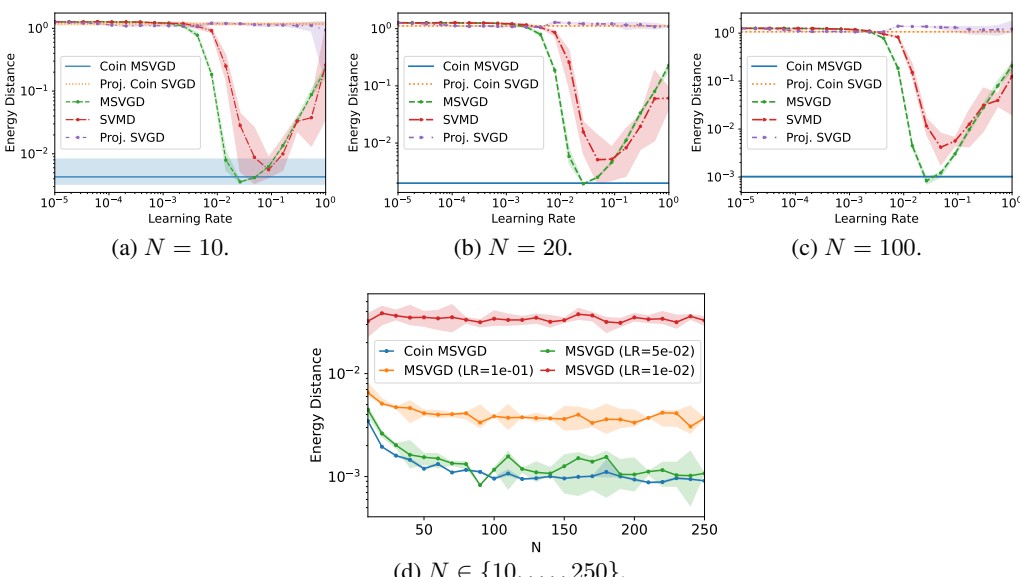

Figure 8: **Additional results for the sparse Dirichlet posterior in [73]**. (a) - (c) Energy distance vs learning rate, for several values of $N$, for Coin MSVGD, MSVGD, SVMD, projected Coin SVGD, and projected SVGD. (d) Energy distance vs $N$, for several different values of the learning rate, for Coin MSVGD and MSVGD. We run each algorithm for $T = 250$ iterations.

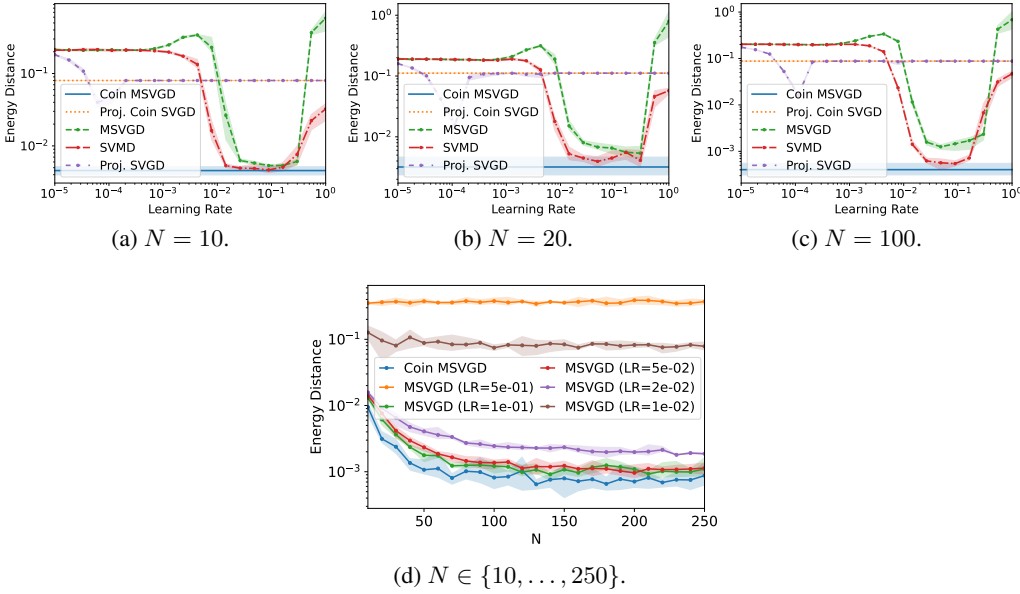

Figure 9: **Additional results for the quadratic target in [1]**. (a) - (c) Energy distance vs learning rate, for several values of $N$, for Coin MSVGD, MSVGD, SVMD, projected Coin SVGD, and projected SVGD. (d) Energy distance vs $N$, for several different values of the learning rate, for Coin MSVGD and MSVGD. We run each algorithm for $T = 500$ iterations.

## G.2 Sampling from a Uniform Distribution

In this section, we consider the task of sampling from the uniform distribution on the unit square, namely $\text{Unif}[-1, 1]^2$. For MSVGD and Coin MSVGD, we once again use the entropic mirror map. We also now use the RBF kernel, with a fixed bandwidth of $h = 0.01$.[4] For MIED and Coin MIED, following [60], we reparameterise the particles using $\tanh$ to eliminate the constraint. We show results for several choices of mollifiers, in each casing choosing the smallest $\epsilon$ which guarantees the optimisation remains stable. We initialise all methods with particles drawn i.i.d. from $\mathcal{U}[-0.5, 0.5]^2$. For both methods, in the interest of a fair comparison, we adapt the learning rates using RMSProp [92]. We use $N = 100$ particles and run each experiment for $T = 250$ iterations. We then assess the quality of the posterior approximations by computing the energy distance to a set of 1000 samples drawn i.i.d. from the true target distribution.

The results for this experiment are shown in Fig. 10. Similar to our previous synthetic examples, Coin MSVGD is competitive with the optimal performance of MSVGD. Meanwhile, for sub-optimal choices of the learning rate, the coin sampling algorithm is clearly preferable. The same is also true for Coin MIED which, depending on the choice of mollifier, obtains comparable or superior performance to MIED with the optimal choice of learning rate.

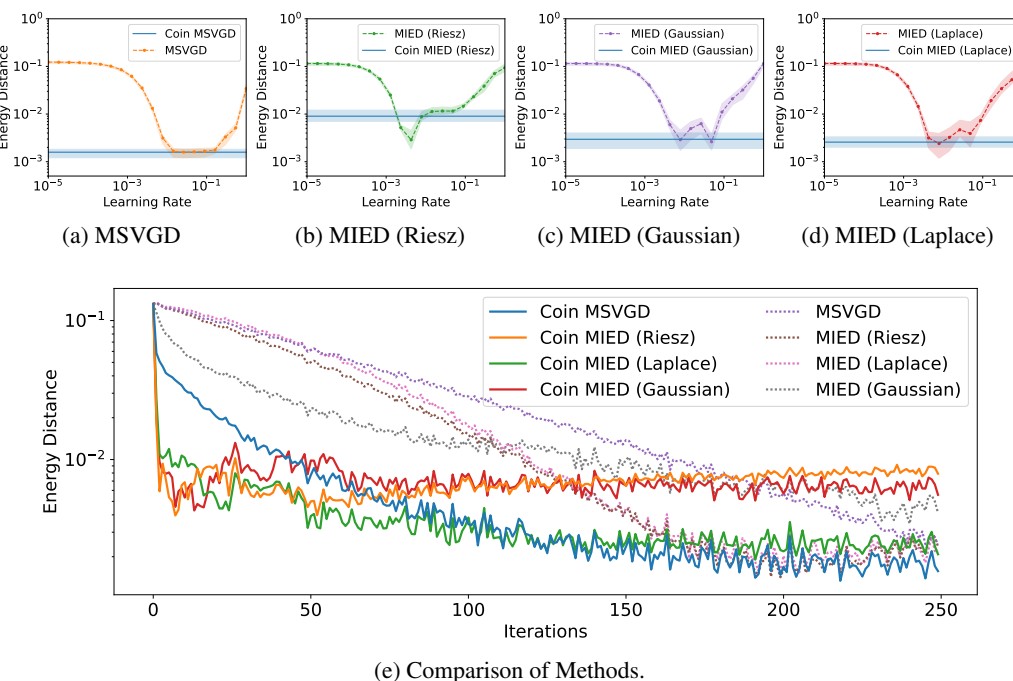

(a) MSVGD    (b) MIED (Riesz)    (c) MIED (Gaussian)    (d) MIED (Laplace)

(e) Comparison of Methods.

Figure 10: **Results for the uniform target:** $\text{Unif}[-1, 1]^2$. Energy distance vs learning rate after 250 iterations for (a) Coin MSVGD and MSVGD; and (b) - (d) Coin MIED and MIED, for three different choices of mollifier.

## G.3 Post-Selection Inference

In this section, we provide additional results for the post-selection inference examples considered in Sec. 6.2. In Fig. 11, we once again plot the coverage of the CIs obtained using Coin MSVGD, MSVGD, SVMD, and the `norejection` algorithm in `selectiveInference` [91], as we vary either the nominal coverage or the total number of samples. Now, however, we consider two sub-optimal choices of the learning rate for MSVGD and SVMD. In particular, we now use an initial learning rate of $\gamma = 0.001$ (Fig. 11a - 11b) or $\gamma = 0.1$ (Fig. 11c - 11d) for RMSProp [92], rather than $\gamma = 0.01$ as in Fig. 3 (see Sec. 6.2).

---

[4]In this example, the use of an adaptive kernel (e.g., with the bandwidth chosen according to the median rule), can cause the samples from MSVGD to collapse [60].

In this case, we see that the performance of both MSVGD and SVMD deteriorates. In particular, they both now obtain significantly lower coverage than Coin MSVGD. As noted in the main text, this increased coverage is of particular importance for smaller sample sizes (see Fig. 11b and Fig. 11c), and for greater nominal coverage levels (see Fig. 11a and 11c), where the standard CIs generally undercover.

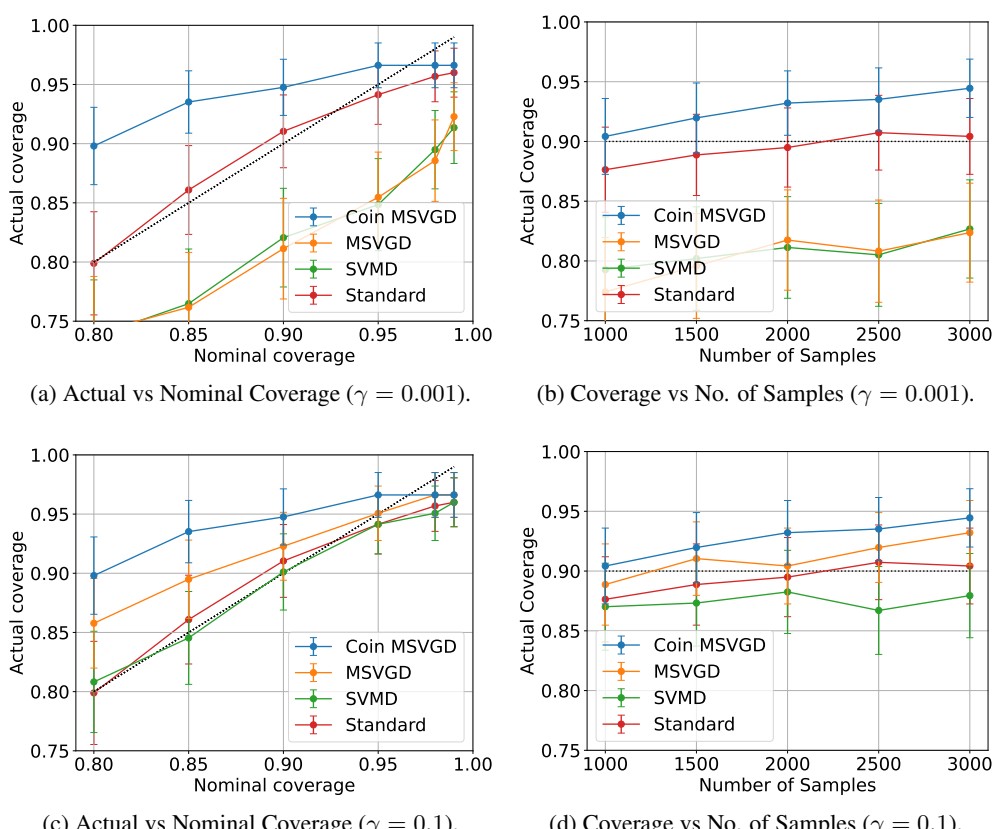

(a) Actual vs Nominal Coverage ($\gamma = 0.001$).

(b) Coverage vs No. of Samples ($\gamma = 0.001$).

(c) Actual vs Nominal Coverage ($\gamma = 0.1$).

(d) Coverage vs No. of Samples ($\gamma = 0.1$).

Figure 11: **Additional coverage results for the post-selection inference.** Coverage of post-selection confidence intervals for Coin MSVGD, MSVGD, and SVMD, using sub-optimal learning rates.

Finally, in Fig. 12, we perform a similar experiment for the real-data example involving the HIV-1 drug resistance dataset. In particular, we compute 90% CIs for the subset of mutations selected by the randomised Lasso, using 5000 samples and the same five methods as before: 'unadjusted' (the unadjusted CIs), 'standard' (the method in `selectiveInference`), MSVGD, SVMD, and Coin MSVGD. Now, however, we use sub-optimal choices of the initial learning rate in RMSProp, namely, $\gamma = 0.001$ (Fig. 12a) and $\gamma = 1.0$ (Fig. 12b), compared to $\gamma = 0.01$ in the main paper.

Similar to the synthetic example, the impact of using a sub-optimal choice of learning rate is clear. In particular, CIs obtained using MSVGD and SVMD now differ substantially from those obtained by Coin MSVGD and the standard approach. This means, in particular, that several mutations are now incorrectly flagged as significant by MSVGD and SVMD (e.g., mutation P41L, P62V, P70R, or P215Y in Fig. 12b), when in reality there is insufficient evidence after conditioning on the selection event.

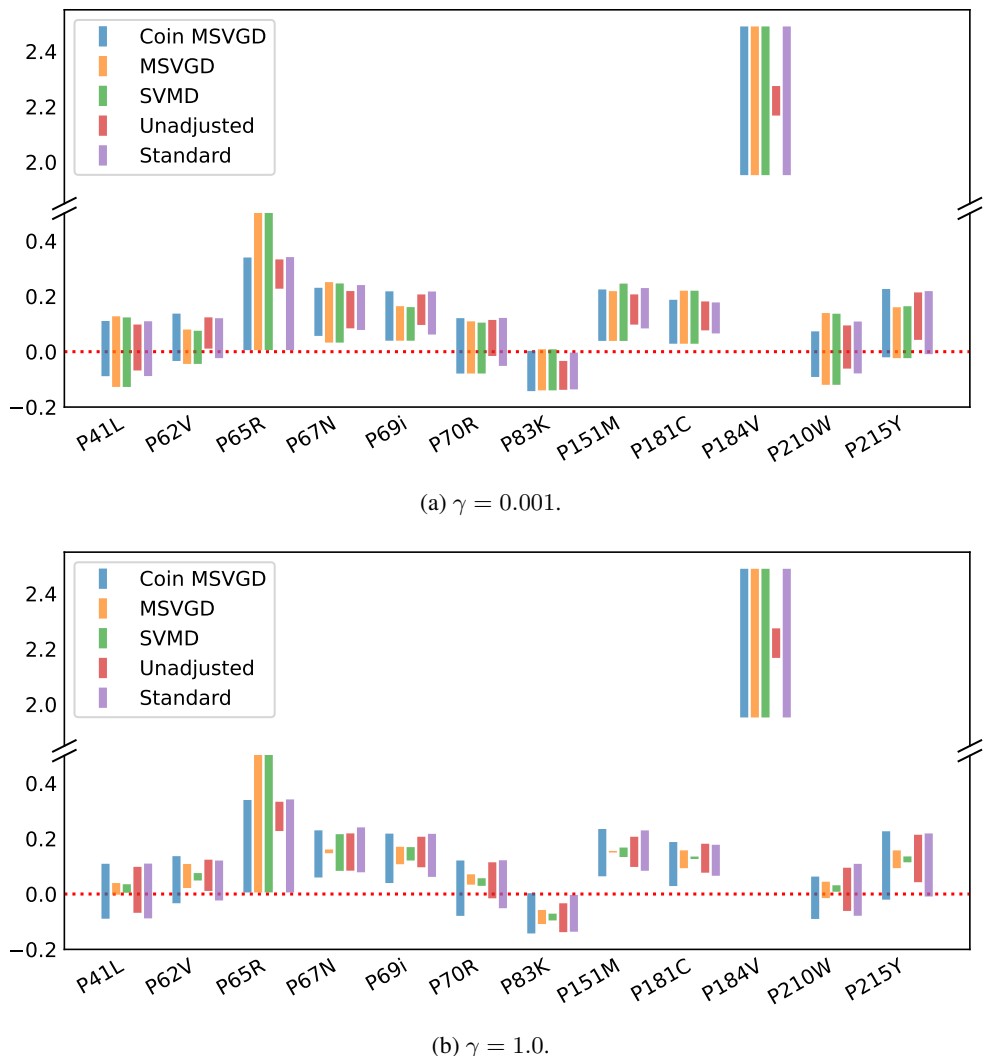

(a) $\gamma = 0.001$.

(b) $\gamma = 1.0$.

Figure 12: **Additional real data results for post-selection inference.** Unadjusted and post-selection confidence intervals for the mutations selected by the randomised Lasso as candidates for HIV-1 drug resistance, for two sub-optimal choices of the learning rate $\gamma$.

