# OpenReview forum: "Learning Rate Free Sampling in Constrained Domains"
_NeurIPS.cc/2023/Conference — NeurIPS 2023 poster_

### Official Review · Reviewer_6R7u · 2023-06-30

**Soundness:** 3 good
**Presentation:** 3 good
**Contribution:** 3 good
**Rating:** 7
**Confidence:** 3

**Summary:**

In this paper the authors first introduce a unified view of some existing sampling algorithms for constrained spaces by exploiting the notion of "mirrored optimisation" from standard convex optimisation in the setting of optimisation on probability spaces.

The problem of sampling is first converted into an optimisation problem through standard arguments of Wasserstein Gradient Flows (WGF) on $\mathbb{R}^d$: 1) consider a minimisation problem $\pi = \mathrm{argmin}_{\mu} \mathcal{F}(\mu)$ for some functional $\mathcal{F}$ and target probability measure $\pi$, 2) find a continuous process which transports from some initial $\eta_0$ to the target $\pi$ via the continuity equation, and c) discretise the initial distribution $\mu_0$ into some $N$ particles. which, when transported along the trajectory $\mu_t$, produces, under suitable assumptions, samples from the target $\pi$, as desired.

To generalise this to possibly constrained spaces $\mathcal{X} \subset \mathbb{R}^d$, the authors then introduce the *Mirrored* Wasserstein Gradient Flows (MWGF) by using the idea of a "mirror map" from convex optimisation. In particular, they use the mirror map to define a bijective transformation from the space of probability measures on the constrained space $\mathcal{P}_2(\mathcal{X})$ to probability measures on unconstrained space $\mathcal{P}_2(\mathbb{R}^d)$.

To make the above construction practically useful, one needs to also discretise the resulting trajectories wrt. time, which then require a choice of stepsize. But equipped with the above view, the authors make use of previous work on "coin sampling", an idea which is based on "coin optimisation" from the convex optimisation literature, to make an explicit choice of stepsize no longer necessary, resulting in a family of particle-based learning rate free sampling methods for constrained domains.


**Strengths:**

There are two main points of the paper that are novel:

1.  The framework of "Mirrored Wasserstein Gradient Flows", though not a huge step from the unconstrained "Wasserstein Gradient Flows", generalises a few previously seen mirrored sampling methods and allows the authors to introduce new mirrored versions of other existing methods for unconstrained spaces.
2.  The combination of "Mirrored Wasserstein Gradient Flows" and "coin sampling", allowing the authors to obtain learning free rate particle-based sampling methods.

(1) is, as far as I can tell, of greatest novelty. Even though instances of this family have previously been seen in the literature, as pointed out by the authors, putting these methods into a more general framework can lead to both improved theoretical understanding of the algorithms and new practically useful methods, the later which the authors nicely demonstrates by the generalisation of existing unconstrained methods to constrained spaces.

The authors nicely demonstrates through empirical results that the introduced methods have empirical advantages over existing methods; in particular, the learning rate free constrained methods perform similarly to their counterparts with finely tuned stepsizes. Given how sensitive some of these algorithms can be to the stepsize, this seems like a useful improvement for practical applications of these methods.

The paper is also very nice to read and not too difficult to follow. The authors also seem to have done a good job mentioning existing works.

**Weaknesses:**

One aspect that might be worth a little bit of questioning is the novelty of the work. As mentioned, the novelty comes from 1) the introduction of a framework capturing several existing mirrored sampling approaches, and 2) the application of coin sampling to the resulting framework. Having seen previous works on mirrored particle-based sampling methods in addition the work on coin sampling, neither of these might seem all too surprising.

With that being said, the authors themselves demonstrate the utility of this formulation by the introduction of two new mirrored sampling methods, and the practical utility of (2) is clearly demonstrated in the empirical section; in total, I think this makes this work more than sufficiently novel.

Another aspect is some lack of clarity regarding the utility of the mirror map in the introduction MWGF in Section 3. Going from the constrained WGF to the unconstrained MWGF is really only using the bijectivity property of the mirror-map $\nabla \phi$; that is, we could easily "generalise" this framework further by just replacing $\nabla \phi$ with *any* bijective map $\psi: \mathcal{X} \to \mathbb{R}^n$, which is commonly what is done under naive application of more classical sampling methods such as HMC to constrained problems (with the additional constraint of differentiability so the push-forward is computable).

I therefore find it somewhat non-obvious as to why the "mirroring" approach is preferable to what I describe above. It seems to me that previous works on mirrored sampling are generally motivated by improved convergence rates in the constrained setting as in Hsieh (2018) and Zhang (2020), which is not going to be the case for any bijective transformation. But in this work there is no mentioning of improved convergence rates of the mirrored versions introduced. I can see why the mirroring is important for the coin sampling, as the proof in Appendix C makes use of fact that the push-forward of strongly log-concave $\pi$ under the mirror map $\nabla \phi$ is also strongly log-concave, but it's so clear why this is imposed already in the MWGF construction. I therefore suspect the authors has taken this less general presentation for the sake of readability, which is completely sensible, but then I'd personally appreciate a minor remark in the main text explaining this and mentioning somewhat clearly why the "restriction" to mirror maps is necessary for the coin sampling.

For the empirical results, one minor weakness is that, as far as I can tell, only a single number of particles is used in each experiment, making one wonder if there is a reason for that choice, or if indeed the observed results generalise nicely for different choices of number of particles. Even though I'd expect the results to generalise, it would be nice to see a few different values for the number of parameters for each experiment rather than one.


**Questions:**

-   As far as I can tell, every experiment is performed using a fixed number of particles ($N = 50$ for some, and $N = 100$ for others). Is this a particular choice, and if so, are there reasons for it, other than computational concerns? Though I would suspect the results to generalise, it would be comforting to see results for a few different choices of $N$.
-   From what I can tell, previous usages of mirror maps in sampling are generally motivated by improving convergence wrt. certain sampler parameters, e.g. stepsize. Do the two algorithms which correspond to mirrored versions of Laplacian adjusted Wasserstein gradient descent and kernel Stein discrepancy descend also posses similar properties? Or is this not something that has been explored yet? (From a quick glance at the appendix, it seem the results are mainly related to continuous-time convergence.)

**Limitations:**

There are two main limitations with the method in its current form: a) since much of the work is based on coin sampling for unconstrained domains, this work suffers from the same ailments, i.e. theoretical results establishing convergence without somewhat strict and non-standard assumptions (which cannot be easily checked in practice) are not yet addressed, and b) any mirrored sampling algorithm requires the availability of a mirror map, and hence so does this work. These limitations are explicitly mentioned by the authors, which is appreciated.

---

> ### Author Rebuttal · Authors · 2023-08-09
>
> We would like to thank the reviewer for their thorough engagement with our work and their constructive feedback. We provide a detailed point by point response to their comments below.
>
> ---
> **Weaknesses**
>
> **One aspect that might be worth a little bit of questioning is the novelty of the work...**
> Many thanks for these considered and thoughtful remarks. We would broadly agree that, given a close familiarity with the constrained sampling literature, as well as the recent work in [1], some of the results in this paper may not be entirely surprising. Nonetheless, as acknowledged by the reviewer, we would highlight several novel contributions.
>
> We introduce the MWGF framework, which elegantly captures many existing constrained sampling approaches, providing a unifying framework for their analysis, as well as allowing us to derive new sampling schemes and analyse their properties. This framework also provides the basis for a principled extension of coin sampling ideas to the constrained setting, resulting in mirrored coin sampling. The result is a highly practical and easy-to-implement algorithm, which consistently obtains state-of-the-art performance. Although not based on the MWGF framework, we also introduce another learning-rate free algorithm, Coin MIED, extending ideas in [2]. Once again, this algorithm achieves excellent performance in empirical testing.
>
> **Another aspect is some lack of clarity regarding the utility of the mirror map...**
> Thanks for raising this point. The reviewer is correct that, for many of the results in App. B, it would be possible to replace the mirror map by a more general bijective map, up to some minor modifications. As the reviewer suggests, our choice is largely one made out of convenience, as well as to aid comparison with existing results. In particular, adopting this formulation makes it easier to contextualise the results in App. B. For example, some of the results we obtain for mirrored LAWGD hold under identical assumptions to those previously used to analyse MSVGD [4].
>
> This aside, we should note that the use of a mirror map does allow us to strengthen several of the results in App. B. In particular, by combining the arguments in [3, App. C], and the results in App. B, we can obtain existential results on the existence of "good mirror maps" which guarantee the convergence of, e.g., MSVGD or MLAWGD to the target at a particular rate, in the spirit of the result for the mirrored Langevin algorithm in [3, Theorem 3], and our own result for mirrored coin sampling [Proposition 16, App. C].
>
> As an example, let us show how to extend [5, Theorem 1] (Theorem 1 in App. B.1) in this way. Suppose we replace [5, Assumption 4] by the assumption that $\pi$ is strongly log-concave. Under this assumption, [3, App. C] guarantees the existence of a mirror map such that the dual target is also strongly log-concave. By the Bakry-Emery criterion, it follows that the $T_p$ inequality is satisfied by the dual target with $p=2$. Thus, [5, Assumption 4] is satisfied and so [5, Theorem 1] holds. In other words, assuming $\pi$ is strongly log-concave, there exists a mirror map such that MSVGD converges to the target at the rate in [5, Theorem 1]. While the assumption that $\pi$ is strongly log-concave is strictly stronger than the original assumption [5, Assumption 4], it is much more tangible, as it relates to the target $\pi$, rather than the dual target $\nu$.
>
> Using similar arguments, we can also extend the other results given in App. B (e.g., Prop. 5 - 7 in App. B.1, Prop. 10 in App. B.2, Prop. 13 - 14 in App. B.3). Given this, we have now added several additional remarks in App. B, summarising how our results can be extended when restricting to a mirror map. As suggested by the reviewer, we have also added an additional remark in Sec 3.2, clarifying our use of a mirror map.
>
> **For the empirical results, one minor weakness is that...only a single number of particles...**
> Thanks for this feedback. The results do generalise across different numbers of particles, but we agree it would be useful to include a comparison in the paper. We have now added an appendix comparing results for different values of $N$, across several experiments.
>
> We include some illustrative results in the attached PDF. In Figs 1(a)-(d), 2(a)-(d), 3(a)-(d), we plot the energy distance as a function of the LR using several different numbers of particles, for the sparse Dirichlet posterior in [6] (Sec 6.1), the quadratic target in [7] (Sec 6.1), and the two-dimensional post-selection inference target [8] (Sec 6.2). In addition, in Figs 1(e), 2(e), 3(e), we plot the energy distance as a function of the number of particles, and for each of the three experiments above.
>
> ---
> **Questions**
>
> **1. As far as I can tell, every experiment is performed...**
> Please refer to our previous response and the attached PDF.
>
> **2. From what I can tell, previous usages of mirror maps in sampling...**
> Thanks for this question. Other than as outlined in our earlier response, we have not yet explored this issue as it relates to mirrored LAWGD and mirrored KSDD. As the reviewer notes, the theoretical results provided in App. B are generally restricted to the continuous time case. However, it would certainly be interesting to the discrete-time properties of these methods further. Given the considerable length of this paper, this is something we feel is best left to future work. Nonetheless, we will add a remark in Sec. 3.2 on this point.
>
> ---
> **References**
>
> [1] L. Sharrock et al. Coin Sampling: Gradient-Based Bayesian Inference without Learning Rates. ICML 2023.
>
> [2] L. Li, et al. Sampling with Mollified Interaction Energy Descent. ICLR 2023.
>
> [3] Y-P Hsieh et al. Mirrored Langevin Dynamics. NeurIPS 2018.
>
> [4] J. Shi et al. Sampling with Mirrored Stein Operators. ICLR 2022.
>
> [5] L. Sun et al. A Note on the Convergence of Mirrored Stein Variational Gradient Descent under (L0,L1) Smoothness Condition. arXiv, 2022.

---

> > ### Comment · Reviewer_6R7u · 2023-08-13
> >
> > I very much appreciate the authors thorough response, and in particular appreciate the discussion regarding extensions and the additional empirical experiments I requested.

---

### Official Review · Reviewer_aGzU · 2023-07-11

**Soundness:** 3 good
**Presentation:** 4 excellent
**Contribution:** 3 good
**Rating:** 6
**Confidence:** 1

**Summary:**

This paper studies constrained sampling with algorithms such as Langevin dynamics or stein-variation gradient. They provide a continuous based framework which can be specialised to various other algorithms as well. The proposed modification of the classical algorithms does not require a well-specified learning rate; instead by utilising a betting scheme the algorithm adjusts its learning rate to maintain a good convergence. A versatile set of numerical examples is provided to support the claim that the iterative methods above converge faster and hence are closer to true distribution than the same instances of the algorithm with badly chosen learning rates.

**Strengths:**

The authors focus on a problem which is often omitted in the development of approximate inference algorithms, namely the choice of learning rate. They focus on constrained settings where they work with continuous versions of mirrored langevin dynamics and stein-variational gradient. The proposed discretized variants of the algorithms. I find these problems of very big practical relevance having encountered these problems myself.

Authors provide a very compelling experimental section where they show the benefit of their betting scheme in improving the convergence rate and/or converged solution. The benchmark problems are very versatile and compare with the right baselines.


**Weaknesses:**

I am not familiar with related work, but it seems a lot of the results are standard from sampling literature with the only novelty being the coin betting which itself is inspired by convex optimization literature works.

There is no theoretical analysis and there is no conjecture even if this works or what are the conditions that this might work, but well this might be a really difficult problem.


**Questions:**

- Can you give an example for operator P from 132?
- Are there other betting strategies which are worth mentioning?
- Does SVMD fall into your framework too? Afaik SVMD is not only-dual algorithm like your proposed Alg. 1? If not why?


**Limitations:**

Tthe authors adequately addressed the limitations and, if applicable, potential negative societal impact of their work

---

> ### Author Rebuttal · Authors · 2023-08-09
>
> We are grateful to the reviewer for their positive remarks, as well as their insightful feedback. We provide a detailed point by point response to their comments below.
>
> ---
> **Weaknesses**
>
> **I am not familiar with related work, but it seems...**
> Thanks for this remark. Although some of the results in App. B do exist in the sampling literature, we would respectfully disagree that the only novelty in this paper is the introduction of learning-rate free algorithms based on coin betting. In particular, we also introduce a general formulation of constrained sampling as a mirrored optimisation problem, and the notion of a MWGF. This provides a unifying framework for existing approaches, and allows us to obtain new constrained sampling algorithms (e.g., MKSDD, MLAWGD) and analyse their convergence. This framework also provides a principled basis from which we derive mirrored coin sampling. In any case, we would argue that introducing several new learning-rate free constrained sampling algorithms is, in itself, a significant and novel contribution, particularly given their impressive numerical performance compared to existing methods.
>
> **There is no theoretical analysis...**
> The reviewer is correct that a theoretical analysis of coin sampling (under standard conditions) is a very difficult question, even in the unconstrained case. We include several remarks on this in Sec 4.2. It is worth emphasising that we do, in fact, provide a discussion of how to obtain a convergence rate for mirrored coin sampling, under an appropriate extension of the conditions in [1], in App. C.
>
> ---
> **Questions**
>
> **1. Can you give an example for operator P...**
>  We could not locate a reference to $P$ in 132. If the question refers to $P_{\mu,k}$ in 133, then this is simply the integral operator $P_{\mu,k}f = \int k(x,\cdot) f(x)\mathrm{d}\mu(x)$. This only differs from $S_{\mu,k}$ in its range.
>
>
> **2. Are there other betting strategies...**
> Thanks for raising this point. While, in this paper, we consider only the KT betting strategy, there are other strategies (e.g., [2]). We now include a remark pointing to this and other relevant references, leaving a more detailed investigation to future work.
>
> **3. Does SVMD fall into your framework too?...**
> Thanks for raising this question. It is currently not clear that SVMD naturally fits into our framework.
>
> As outlined in the paper, MSVGD can naturally be viewed as a kernelised version of the WGF of the KL w.r.t. the dual target in the dual space. When only a single particle is used, MSVGD reduces to gradient descent w.r.t. the negative log-density of the dual target [3]. In this sense, MSVGD can be viewed as the SVGD-analogue of the mirrored Langevin dynamics in [4]. In particular, [4] proposes running the unadjusted Langevin algorithm w.r.t. the dual target. Thus, if one removes the noise, the scheme in [4] also reduces to gradient descent w.r.t. the negative log-density of the dual target. On the other hand, we would argue that SVMD is best viewed as a kernelised approximation to the Wasserstein mirror flow w.r.t. the (primal) target [5, App. C]. In particular, with a single particle, SVMD reduces to mirror descent w.r.t. the negative log-density of the (primal) target [3, Sec 4.3]. In this sense, SVMD can be viewed as the SVGD-analogue of the mirror Langevin diffusion in [5]. Indeed, when the noise is removed in [5], this scheme also reduces to mirror descent w.r.t. the negative log-density of the (primal) target.
>
> Given this interpretation, a natural question is then whether one can obtain a coin-sampling analogue of SVMD. In principle, one can certainly write down and implement "Coin SVMD". However, given that SVMD does not naturally fit into the MWGF framework, as outlined above, it is unclear that this is a particularly well-justified approach. Generally speaking, coin sampling analogues of ParVI algorithms perform well when coin sampling is used in place of a time-discretisation of a standard WGF. In such cases, when using a single particle, existing ParVI algorithms reduce to gradient descent, and coin sampling reduces to the coin betting algorithm [6], in either case w.r.t the negative log-density of the target. As outlined above, this is essentially the framework for MSVGD and Coin MSVGD, although now the updates take place in the dual space w.r.t. the dual target. In particular, if one uses a single particle, MSVGD and Coin MSVGD reduce to gradient descent or coin betting w.r.t. the negative log-density of the dual target. On the other hand, it is not clear that coin sampling can be used in cases where existing algorithms correspond to time-discretisations of `non-standard' WGFs, e.g., the Wasserstein mirror flow [5, App. C]. As noted above, in these cases, when one uses a single particle (SVMD), or removes the noise term (mirror Langevin diffusion), these algorithms reduce to mirror descent w.r.t. the negative log-density of the (primal) target.
>
> This hypothesis is partially based on some preliminary numerical experiments. Indeed, after implementing "Coin SVMD", we observed numerical instabilities across various experiments. This is in contrast to Coin MSVGD, which performed consistently across experiments, and was derived in a principled way based on the MWGF framework. While beyond the scope of this work, we feel that a detailed investigation of these issues would be an interesting avenue for future work.
>
> ---
> **References**
>
> [1] L. Sharrock et al. Coin Sampling: Gradient-Based Bayesian Inference without Learning Rates. ICML 2023.
>
> [2] A. Cutkosky et al. Black-Box Reductions for Parameter-free Online Learning in Banach Spaces. COLT 2018.
>
> [3] J. Shi et al. Sampling with Mirrored Stein Operators. ICLR 2022.
>
> [4] Y-P Hsieh et al. Mirrored Langevin Dynamics. NeurIPS 2018.
>
> [5] K. Ahn et al. Efficient constrained sampling via the mirror-Langevin algorithm. NeurIPS 2021.
>
> [6] F. Orabona et al. Coin Betting and Parameter-Free Online Learning. NeurIPS 2016.

---

> > ### Comment · Reviewer_aGzU · 2023-08-14
> > **response**
> >
> > Thank you for your response. As you can see from my score, this is not my field, but your paper was understandable, clearly written, and the problem studied was indeed important for which not many off-the-shelf solutions are available. For other aspects I let other reviewers decide.

---

### Official Review · Reviewer_Uc6g · 2023-07-25

**Soundness:** 3 good
**Presentation:** 3 good
**Contribution:** 3 good
**Rating:** 7
**Confidence:** 3

**Summary:**

The problem of sampling from unnormalised probability distributions is of central importance to computational statistics and machine learning. The well-known SVGD method appears to break down when applied to constrained targets. Other recent methods share the same limitation, namely, significantly depend on a suitable choice of learning rate.

Hence, this paper mainly focus on constrained sampling algorithms. The authors propose a suite of particle-based algorithms (coin MSVGD, coin MIED), which are entirely learning rate free.

Detailed theoretical description and justification are provided in the paper. Several numerical experiments (simplex targets, post selection inference, and fairness bayesian neural network) are also conducted to evaluate the performance of proposed methods, superior or competitive results are obtained when compared to other methods.


**Strengths:**

1. The idea of viewing constrained sampling as a mirrored optimisation problem is interesting.
2. A suite of learning rate free methods is provided, which will facilite researchers from different disciplines that related to sampling from unnormalised probability distributions.
3. The paper is well written and easy to read.

**Weaknesses:**

1. Analysis of the results is not enough.
2. On the second numerical experiment, advantage of this method is marginal.

**Questions:**

I have the some minor questions/comments for the authors:

1. In the result of Section 6.1 (Figure 1), although MSVGD and SVMD are sensitive to learning rate, SVMD seems to converge at lower energy distance when using a proper learning rate (eg. 1e-1). I wonder the reason of this phenomenon, and is there any solutions to achieve similar performance for Coin MSVGD?

2. Also in Figure 1, when learning rate is higher than 1e-3, Projected SVGD is also learning rate free, so does Coin MSVGD has any other advantage over it?

3. The presentation of Figure 3 is not clear and straight enough to compare the result difference of the four methods.

4. Move some method background and detailed algorithm description to Appendix and add more result analysis in the main paper.

5. The proposed method is based on [76] and [56], it would be better to describe the connection and difference more clearly in the paper.

6. First sentence of second paragraph in Introduction: 'While such methods have enjoyed great success in sampling from unconstrained distributions, they typically break down when applied to constrained targets', may needs a reference.

**Limitations:**

The authors have highlighted two limitations of their work, and the limitations are still open problems in the research domain.

---

> ### Author Rebuttal · Authors · 2023-08-09
>
> We are grateful to the reviewer for their constructive feedback. We provide a detailed point by point response to their comments below.
>
> ---
> **Weaknesses**
>
> **Analysis of the results is not enough.**
> Thanks for this feedback. As noted in Sec 4.2 and discussed in [1], establishing the theoretical properties of coin sampling methods under standard conditions is very challenging, and remains unresolved even in the unconstrained case. This being said, we do actually discuss how to obtain a convergence rate for mirrored coin sampling in App. C, based on an extension of [1]. We also provide a detailed analysis of the convergence properties of MWGFs in continuous-time in App. B.
>
> **On the second numerical experiment...**
> We agree that the advantage of our method in this experiment may seem marginal. We would, however, note the following. First, the optimal learning rate for, e.g., MSVGD, is not known a priori, and must therefore be tuned by hand. This comes at the cost of an additional computational expense. Second, if the learning rate for MSVGD is poorly tuned, Coin MSVGD does lead to a clear advantage (Fig. 9 in App. G.3). Third, empirically, Coin MSVGD seems to converge much faster than MSVGD (Fig. 8 in App G.3), even when using a well tuned learning rate. Although one can obtain comparable results by using a larger learning rate for MSVGD, this can lead to non-convergence (Fig. 8 in App. G.3). These points could be emphasised more clearly in the main text, which we have now addressed.
>
> ---
> **Questions**
>
> **1. In the result of Sec 6.1...**
> Thanks for this question. There is, indeed, a good reason why SVMD compares favourably to (Coin) MSVGD in this example, which we explain below. Before we do so, we should note that Fig. 1a and 1b were labelled in reverse in the original submission (this has now been corrected). With this said, let us consider the results for the quadratic target (Fig. 1a). In this case, the target is log-concave, while the dual target is not. In such cases, SVMD is expected to outperform (Coin) MSVGD, as it can exploit log-concavity in the primal space. This is further discussed in App. G.1.
>
> Currently, it is unclear whether it is possible to obtain a coin sampling algorithm that can obtain comparable results in this case. While, in principle, one can write down a coin sampling analogue of SVMD, it is not evident that this is a principled approach.
>
> Generally, coin sampling analogues of ParVI algorithms perform well when the coin sampling updates are used in place of a time-discretisation of a standard WGF. In such cases, when using a single particle, existing algorithms reduce to gradient descent w.r.t. the negative log-density of the target, and coin sampling reduces to coin betting [2]. This is essentially the framework for MSVGD and Coin MSVGD, even though the updates are taking place in the dual space. In particular, with a single particle, MSVGD and Coin MSVGD reduce to gradient descent or coin betting w.r.t. the negative log-density of the dual target.
>
> On the other hand, it is not clear coin sampling can be used when existing algorithms correspond to a `non-standard' WGF, e.g., the Wasserstein mirror flow [3], which yields the mirror Langevin diffusion [3] and SVMD [4]. In these cases, if one uses a single particle (SVMD), or removes the noise (mirror Langevin diffusion), these algorithms reduce to a mirror flow (i.e., Riemannian gradient flow) w.r.t. the negative log-density of the (primal) target.
>
> While beyond the scope of this paper, we believe further study of this topic would be a very interesting avenue for future work.
>
> **2. Also in Fig. 1, when learning rate is...**
> As noted in Sec. 6, projected SVGD is just SVGD with a Euclidean projection onto $\mathcal{X}$ after each update. As such, projected SVGD does always depends on a learning rate, even though in this example the results for projected SVGD with LRs greater than $1\times 10^{-3}$ may appear to be identical. In terms of other advantages of Coin MSVGD over projected SVGD in this example, we would highlight that Coin MSVGD converges to the target distribution, while projected SVGD fails to converge for any value of the learning rate.
>
> **3. The presentation of Fig. 3 is not clear...**
> Thanks for raising this concern. We have now added additional remarks describing this figure in more detail, which should help to aid comparison between the various methods.
>
> **4. Move some method background...**
> Thanks for this feedback. Given that coin sampling is a rather new approach, we feel it is important to include a detailed description of our methodology and algorithms in the main paper. This being said, we agree that some readers may appreciate seeing some more results on MWGFs in the main text. As such, we have now included in Sec 3.2 a much more detailed description of the convergence results in App. B. If space allows, we will also include a basic dissipation result for the MWGF (4).
>
> **5. The proposed method is based on [76] and [56]...**
> We also agree that a more explicit delineation between our approaches and [56, 76] would be appreciated. We have therefore added additional remarks in Sections 4.2 and 4.3, clarifying the relationships between Coin MSVGD and Coin SVGD, and between Coin MIED and MIED, respectively. We have also added an additional appendix providing a more detailed overview of MIED and its relationship to other ParVI methods.
>
> **6. First sentence of second paragraph in Introduction...**
> We have added references to [4, 5] after this sentence.
>
> ---
> **References**
>
> [1] L. Sharrock et al. Coin Sampling: Gradient-Based Bayesian Inference without Learning Rates. ICML 2023.
>
> [2] F. Orabona et al. Coin Betting and Parameter-Free Online Learning. NeurIPS 2016.
>
> [3] K. Ahn et al. Efficient constrained sampling via the mirror-Langevin algorithm. NeurIPS 2021.
>
> [4] J. Shi et al. Sampling with Mirrored Stein Operators. ICLR 2022.
>
> [5] Y.P. Hsieh et al. Mirrored Langevin Dynamics. NeurIPS 2018.

---

> > ### Comment · Reviewer_Uc6g · 2023-08-11
> >
> > Thanks for authors' detailed explanation.
> >
> > The authors have addressed all my questions and have demonstrated them with additional experiments. Based on the response and comments from other reviewers, I tend to keep my current evaluation.

---

### Official Review · Reviewer_fTFt · 2023-07-25

**Soundness:** 3 good
**Presentation:** 3 good
**Contribution:** 3 good
**Rating:** 8
**Confidence:** 3

**Summary:**

The authors derive a general class of solutions to the constrained sampling problem via MWGF, which incorporates various existing constrained sampling techniques. They do this by viewing sampling in constrained domains as a mirrored optimization problem on the space of probability measures.
The paper presents a set of new particle-based sampling algorithms for constrained domains that are entirely learning rate free, building on the recently introduced coin sampling methodology. The efficacy of these algorithms is demonstrated on various numerical examples.

**Strengths:**

Despite the availability of various efficient methods for sampling from unconstrained distributions, which have demonstrated success across numerous applications, the effectiveness of analogous techniques for constrained targets largely relies on selecting an appropriate learning rate. The findings presented in this paper demonstrate that the suggested learning-rate free algorithms achieve comparable performance to finely tuned constrained sampling methods, eliminating the need for hyperparameter tuning. Moreover, the paper thoroughly considers related research and emphasizes the originality and advantages of the proposed approach.

**Weaknesses:**

The authors do not address the computational complexity of their methods, which would be beneficial for comparing them with other approaches. Additionally, it is somewhat cumbersome to switch back and forth between the main body and the appendix to comprehend the results in Section 6.

**Questions:**

1) Line 74: Is the assumption of "uniquely minimized" generally valid?

2) Line 197: typo, "using" instead of "sing"

3) Line 256: Why did you chose the IMQ kernel in this case?

4) Figure 4: Are the legends in (b) and (c) accurate? Shouldn't the time increase?

**Limitations:**

The authors adequately addressed the limitations of their approach in a dedicated paragraph (see Section 7).

---

> ### Author Rebuttal · Authors · 2023-08-09
>
> We would like to thank the reviewer for their positive comments and constructive feedback. We provide a point by point response to their comments below.
>
> ---
>
> **Weaknesses**
>
> **The authors do not address the computational complexity...**
> Thanks for this comment. We do, in fact, provide a discussion of the computational complexity of our algorithm in App. E (lines 941 - 948). To summarise, the time complexity of both Coin MSVGD and Coin MIED is $O(N^2)$ per iteration, which is identical to MSVGD and MIED. In experiments, we also found that our algorithms took essentially the same time as MSVGD and MIED. However, we acknowledge that this discussion wasn't signposted in the main body of the paper, so it was easy to miss. We will update this in the camera-ready version.
>
> **It is somewhat cumbersome to switch back and forth...**
> Thanks for highlighting this. Using the additional page allowed in the camera ready version, we plan to move several Figs from the appendices to the main body of the paper, which should help to include readability. Using this additional page, we will also add additional detail wherever we have referred to results given in the appendices, ensuring as far as possible to make the description of any results in the appendices self contained.
>
> ---
> **Questions**
>
> **1. Line 74: Is the assumption of "uniquely minimized" generally valid?**
> This assumption is indeed valid for the dissimilarity functionals considered in this paper, as well as those considered more widely in the literature. In particular, this is true for $\mathcal{F}(\mu) = \mathrm{KL}(\mu|\pi)$, as well as for other $f$-divergences such as the $\chi^2$-divergence. Under mild assumptions (e.g., use of a characteristic kernel), this is also the case for other dissimilarity functionals whose Wasserstein gradient flows have been studied in the literature, including the MMD [1] and the KSD [2]. We refer to, e.g., Theorem 5 in [3] for (mild) assumptions under which the MMD uniquely vanishes at the $\mu=\pi$.
>
> **2. Line 197: typo, "using" instead of "sing".**
> Thanks for spotting this typo, it has now been corrected.
>
> **3. Line 256: Why did you chose the IMQ kernel in this case?**
> We use the IMQ kernel here due to its convergence control properties. In particular, the IMQ kernel is known to metrize weak convergence [4, Theorem 8]. While this was previously mentioned in the appendix on 'Additional Numerical Details' (App. F), we will make sure to also mention this in the main text.
>
> **4. Fig. 4: Are the legends in (b) and (c) accurate? Shouldn't the time increase?**
> The legends in Fig. 4(b) and 4(c) are, indeed, accurate. It is worth noting that $t$ denotes the value of the fairness constraint here, rather than the time. We follow the standard notation here, see, e.g., [5], although we acknowledge that this notation may be slightly confusing. We have now added an additional remark in the caption to clarify the meaning of $t$.
>
> ---
>
> **References**
>
> [1] Michael Arbel et al. Maximum Mean Discrepancy Gradient Flow. NeurIPS 2019.
>
> [2] Anna Korba et al. Kernel Stein Discrepancy Descent. ICML 2021.
>
> [3] Arthur Gretton, Karsten M. Borgwardt, Malte J. Rasch, Bernhard Scholkopf, Alexander Smola. A Kernel Two-Sample Test. JMLR, 13, 2012.
>
> [4] Jackson Gorham and Lester Mackey. Measuring sample quality with kernels. ICML 2017.
>
> [5] Lingxiao Li, Qiang Liu, Anna Korba, Mikhail Yurochkin, and Justin Solomon. Sampling with Mollified Interaction Energy Descent. ICLR 2023.

---

### Author Rebuttal · Authors · 2023-08-09

Many thanks to all of the reviewers for their thorough engagement with our work, and for the many constructive comments. We have made several revisions to our original submission in response to the reviewers' feedback, which we feel have further improved our paper. Full details are provided in our individual responses to each of the reviewers below.

At the request of reviewer 6R7u, we have also performed some additional numerical experiments, verifying that our results generalise for different numbers of particles. We report the results of these experiments in an additional appendix in the latest revision of our paper. We also provide a sample of these results in the attached PDF.

---

### Decision · Program_Chairs · 2023-09-21

**Decision:**

Accept (poster)

**Comment:**

This paper considers the problem of sampling in constrained spaces. Many widely-studied algorithms like Langevin dynamics require learning rates (unlike in unconstrained spaces). This paper proposes a set of particle-based methods that rely on coin betting rather than learning rates. Reviewers agreed this was an important problem and useful solution.